# Model-Free Learning for Two-Player Zero-Sum Partially Observable Markov Games with Perfect Recall

**Tadashi Kozuno**[*]
University of Alberta
tadashi.kozuno@gmail.com

**Pierre Ménard**[*]
Otto von Guericke Universität Magdeburg
pierre.menard@ovgu.de

**Rémi Munos**
DeepMind Paris
munos@deepmind.com

**Michal Valko**
DeepMind Paris
valkom@deepmind.com

## Abstract

We study the problem of learning a Nash equilibrium (NE) in an imperfect information game (IIG) through self-play. Precisely, we focus on two-player, zero-sum, episodic, tabular IIG under the *perfect-recall* assumption where the only feedback is realizations of the game (bandit feedback). In particular, the *dynamics of the IIG is not known*—we can only access it by sampling or interacting with a game simulator. For this learning setting, we provide the Implicit Exploration Online Mirror Descent (IXOMD) algorithm. It is a model-free algorithm with a high-probability bound on the convergence rate to the NE of order $1/\sqrt{T}$ where $T$ is the number of played games. Moreover, IXOMD is computationally efficient as it needs to perform the *updates only along the sampled trajectory*.

## 1  Introduction

We study the setting of *learning* a Nash equilibrium (NE, Nash Jr, 1950) in an *imperfect information game* (IIG, Osborne and Rubinstein, 1994). Precisely, we focus on two-player zero-sum IIG under the *perfect-recall* assumption (Kuhn, 1953). Perfect recall means that the players *do not forget* observations encountered or actions taken during the game. We model the game as a tabular, episodic of horizon $H$, *partially observable Markov game* (POMG) with a state space of size $S$, action spaces of size $A$ and $B$ for the max- and min-player respectively, and observation spaces (i.e., information set spaces, which are partitions of the state space) of size $X$ and $Y$ for the max- and min-player. In learning by *self play*, we control *both* the max *and* min-player. After $T$ episodes of the game we are asked to return a profile that is close to a NE in terms of *exploitability* gap (Ponsen et al., 2011).

**Full feedback**   In case when we have perfect knowledge of the game (i.e., the transition probabilities and rewards) there already exist several methods approximating the NE. The first line of work casts the setting through the sequence-form representation as a linear program which can be solved efficiently for games with moderate sizes of observation spaces $X$ and $Y$ (Romanovsky, 1962; von Stengel, 1996; Koller et al., 1996). The sequence-from representation allows also to cast the setting as *finding a saddle point* (Hoda et al., 2010). It is then possible to adapt first-order methods such as Nesterov's smoothing (Nesterov, 2005) and MirrorProx (Nemirovski, 2004) to IIG, as done respectively by Hoda et al. (2010); Kroer et al. (2018) and Kroer et al. (2015, 2020). These methods have a rate of

---

[*]Equal contribution

convergence of order $\widetilde{\mathcal{O}}((X+Y)/T)$, where $\widetilde{\mathcal{O}}$ hides poly-log terms in $e^H, X, A, Y, B, T$.[2] Note that *game-dependent* exponential rate could also be obtained with first-order methods, see Gilpin et al. (2012) and Munos et al. (2020). Another important line of work relies on minimizing the *counterfactual regret* (Zinkevich et al., 2007). It uses an algorithm designed for adversarial bandits to locally minimize the regret of each player. A well-known example is CFR by Zinkevich et al. (2007) based on the regret-matching algorithm (Hart and Mas-Colell, 2000; Gordon, 2007). There exist many other variants of it, such as CFR+ (Tammelin, 2014; Burch et al., 2019), see also Farina et al. (2019, 2021a). These algorithms however only enjoy a (known) guarantee of convergence of order $\widetilde{\mathcal{O}}((X\sqrt{A} + Y\sqrt{B})/\sqrt{T})$. Note that the two last approaches require computing a full feedback: either some gradient for the first-order methods or the local regret for counterfactual regret minimization. Usually, this can be done by a complete traversal of the state space leading to a time-complexity of order $\mathcal{O}(S)$. Sampling can reduce this time-complexity to $\mathcal{O}(X+Y)$,[3] i.e., we sample the transitions and the actions of the other player; see for example the external-sampling MCCFR algorithm (Lanctot et al., 2009; Farina et al., 2020).

**Bandit feedback**    In this paper, we consider a more challenging setting where we *only observe realizations of the games (bandit feedback) and do not have any prior knowledge of the game*. Precisely, the rewards, the transition probabilities (sometimes modeled as the policy of a *chance player*), the observation/state space, and its (tree) structure are unknown.

**Bandit feedback, model-based**    To deal with the limited bandit feedback, Zhou et al. (2020) consider model-based approach by using *posterior sampling* (PS, Strens, 2000) to learn a model and then use the CFR algorithm in games sampled from the posterior. They obtain a convergence rate of order $\widetilde{\mathcal{O}}(\max(XA + YB, \sqrt{S})/\sqrt{T})$ but only when the games are actually sampled according to the known prior. In addition, they still need to know the state space and its structure[4] in order to instantiate the prior. Instead, Zhang and Sandholm (2021) rely on the principle of optimism in presence of uncertainty to incrementally build a model of the game. Then, they feed optimistic local regrets to a counterfactual regret minimizer algorithm such as the CFR algorithm. They prove a high-probability bound on the exploitability gap of order $\widetilde{\mathcal{O}}((X\sqrt{A} + Y\sqrt{B})/\sqrt{T})$.

**Bandit feedback, model-free**    Our results follows another line of work which consider a *model-free* approach. A well known algorithm of this type is outcome-sampling MCCFR (Lanctot et al., 2009; Farina et al., 2020), which builds an importance-sampling estimate of the counterfactual regret given *exploration profile* (named balanced strategy by Farina et al., 2020). This exploration strategy should ensure that the players explore the information sets uniformly (i.e., such that all induced reach probabilities are lower-bounded by an absolute constant). Note that it is not clear how to find such an exploration profile without knowing the structure of the game.[4] In particular, following the uniform distribution over the actions at each information set is not necessarily a good choice, e.g., when the tree formed by the information set space is not balanced. This algorithm has a guarantee of order $\widetilde{\mathcal{O}}((X\sqrt{A} + Y\sqrt{B})/\sqrt{T})$ with high probability. Building on this idea, Farina and Sandholm (2021) propose to mix the exploration profile with one produced by a counterfactual regret minimizer such as CFR. They prove a high-probability bound on the exploitability gap of order $\widetilde{\mathcal{O}}(\text{poly}(X, A, Y, B)/T^{1/4})$. Note that this bound is a consequence of a bound on the regret of both players (see Section 2) that holds even in the non-stochastic setting where an adversary picks a new game at each episode. Closer to our approach, Farina et al. (2021b) recast the setting to an adversarial bandit linear optimization (Flaxman et al., 2005; Abernethy et al., 2008, see also Section 3.1). Precisely, they use the online mirror descent (OMD) algorithm with the dilated entropy distance-generating function (Hoda et al., 2010; Kroer et al., 2015) as regularizer. Then, OMD is fed with an estimate of the losses of the reformulated adversarial bandit linear instance. The estimator is a generalization of the typical one-point linear regression (Dani et al., 2008). They obtain a rate of order $\widetilde{\mathcal{O}}((XA + YB)/\sqrt{T})$, which is, similarly as done by Farina and Sandholm (2021), derived from a regret bound valid in the adversarial setting. However, their bound holds only in expectation and not in high probability.

---

[2]Therefore, we hide *polynomial* dependence on the horizon $H$.

[3]Note that $\mathcal{O}(X+Y)$ is at most $\mathcal{O}(S)$.

[4]By *structure* we refer to the tree structure of the state space or observations spaces, see Section 2.

| Algorithm | | Adv. game | Rate |
|---|---|---|---|
| Zhou et al. (2020) | model-based | no | $\widetilde{\mathcal{O}}(\max(X\sqrt{A}+Y\sqrt{B}, \sqrt{S})/\sqrt{T})$ [1] |
| Zhang and Sandholm (2021) | | | $\widetilde{\mathcal{O}}((X\sqrt{A}+Y\sqrt{B})/\sqrt{T})$ |
| Lanctot et al. (2009); Farina et al. (2020) | model-free | yes | $\widetilde{\mathcal{O}}((X\sqrt{A}+Y\sqrt{B})/\sqrt{T})$ |
| Farina and Sandholm (2021) | | | $\widetilde{\mathcal{O}}(\mathrm{poly}(X,A,Y,B)/T^{1/4})$ |
| Farina et al. (2021b) | | | $\widetilde{\mathcal{O}}((XA+YB)/\sqrt{T})$ [2] |
| IXOMD (this paper) | | | $\widetilde{\mathcal{O}}((X\sqrt{A}+Y\sqrt{B})/\sqrt{T})$ |

Table 1: Algorithms for computing a NE of an IIG with bandit feedback and their respective upper bound on the exploitability gap after $T$ episodes. In the adversarial game column we precise whether the algorithm could be used to obtain a $\sqrt{T}$-regret for one player when the other player and the game are chosen by an adversary at each episodes.

[1] Only in expectation according to a known prior on the game.
[2] Only in expectation.

To obtain high-probability bound, we instead propose to use an importance sampling estimator of the losses with *implicit exploration* (Kocák et al., 2014; Neu, 2015). Indeed, the implicit bias of this estimator allows to effortlessly control the variance of the estimate, see Lattimore and Szepesvári (2020, Chapter 12) for an in-depth discussion. Using this estimator, we give the Implicit Exploration Online Mirror Descent (IXOMD) based on OMD with the dilated entropy distance-generating function (using uniform weights) as a regularizer and add implicit exploration in the importance sampling estimator of the losses. Using our new analysis of this particular combination, we prove a high-probability bound on the exploitability gap of the average profile of order $\widetilde{\mathcal{O}}((X\sqrt{A}+Y\sqrt{B})/\sqrt{T})$; cf. Table 1 to see how our result compares to the prior work mentioned above. Precisely, our bound is obtained by bounding the regret of each player if they both follow the policy prescribed by IXOMD. Note that the regret bound, e.g., of the max-player, of order $\widetilde{\mathcal{O}}(X\sqrt{AT})$, remains valid if the opponent's policy *and* the game are picked by an adversary at each episode. IXOMD shares some similarities with the approach of Jin et al. (2020) designed for a different setting (see Remark 1). A notable difference is that we use the dilated entropy distance-generating function as a regularizer instead of the un-normalized Kullback-Leibler divergence (Rosenberg and Mansour, 2019). Our choice of regularizer allows an efficient update of the current policy with a $\mathcal{O}(HA)$ time-complexity per episode (see Section 3.3). In particular, our result answers the open problem raised by Farina et al. (2021b) and Farina and Sandholm (2021) of providing an algorithm with high-probability regret bound scaling with $\sqrt{T}$ with $\mathcal{O}(HA)$ computations per episode. Interestingly, we can also update the average profile (which will be returned at the end of the learning, see Section 3.3) in an online fashion. As consequence, IXOMD enjoys an overall time-complexity of $\mathcal{O}(TH(A+B) + \min(TH, X)A + \min(TH, Y)B)$ and space-complexity of order $\mathcal{O}(\min(TH, X)A + \min(TH, Y)B)$.

Moreover, IXOMD requires almost no prior knowledge of the game. In particular, we do not need to know the list of information sets in advance. We only require an oracle providing the possible actions at encountered information sets and a bound on $A$, $B$, and $H$ to optimally[5] tune the learning rate, see Remark 4.

We highlight our main contributions:

- We give the IXOMD algorithm that learns a NE of an IIG in self-play with limited feedback. It has a provably high-probability convergence rate of order $\widetilde{\mathcal{O}}((X\sqrt{A}+Y\sqrt{B})/\sqrt{T})$. The time-complexity of IXOMD is of order $\mathcal{O}(TH(A+B) + \min(TH, X)A + \min(TH, Y)B)$ with a space-complexity of order $\mathcal{O}(\min(TH, X)A + \min(TH, Y)B)$.

- If only one player follows IXOMD, e.g., the max-player, then its regret is w.h.p. at most $\widetilde{\mathcal{O}}(X\sqrt{AT})$. The important property of our result is that it remains valid even if the policy *and* the game are picked by an adversary at each episode. Furthermore, the time-complexity

[5] Precisely, with this knowledge we obtain a regret bound, e.g. for the max-player, of order $\widetilde{\mathcal{O}}(X\sqrt{AT})$; whereas we get $\widetilde{\mathcal{O}}(XA\sqrt{T})$ without it.

of `IXOMD` per episode is of order $\mathcal{O}(HA)$. This answers an open problem of Farina et al. (2021b); Farina and Sandholm (2021).

- `IXOMD` only needs to know the possible actions at the encountered information sets and a bound on $A$, $B$, and $H$ to tune the learning rate. In particular, we do not need to know the list of information sets in advance.

## 2 Preliminaries

In this section, we introduce our notations and our setting—partially observable Markov game (POMG) with bandit feedback and perfect recall. For a positive integer $i$, we denote by $[i]$ the set $\{1, 2, \ldots, i\}$. For a finite set $\mathcal{A}$, we let $\Delta_{\mathcal{A}}$ or $\Delta(\mathcal{A})$ denote the set of all probability distributions over $\mathcal{A}$.

**Partially observable Markov game (POMG)**   We consider an episodic, tabular, two-player, zero-sum POMG $(\mathcal{S}, \mathcal{X}, \mathcal{Y}, \mathcal{A}, \mathcal{B}, H, \{p_h\}_{h \in [H]}, \{r_h\}_{h \in [H]})$, which consists of the following components (Littman, 1994; Shapley, 1953): a finite state space $\mathcal{S}$ of size $S$, its information set spaces (partitions of $\mathcal{S}$) $\mathcal{X}$ of size $X$ and $\mathcal{Y}$ of size $Y$ for the max- and min-player (resp.), finite action spaces $\mathcal{A}$ of size $A$ and $\mathcal{B}$ of size $B$ for the max- and min-player (resp.), time-horizon $H \in \mathbb{N}$, initial state distribution $p_0 \in \Delta(\mathcal{S})$, a state-transition probability kernel $p_h : \mathcal{S} \times \mathcal{A} \times \mathcal{B} \to \Delta(\mathcal{S})$ for each $h \in [H]$, and a reward function $r_h : \mathcal{S} \times \mathcal{A} \times \mathcal{B} \to [0, 1]$ for each $h \in [H]$. For a state $s \in \mathcal{S}$ we denote by $x(s) \in \mathcal{X}$ and $y(s) \in \mathcal{Y}$ information sets such that $s \in x(s)$ and $y \in y(s)$.

**Learning procedure**   The players play this game for $T$ episodes, following so-called policies. A policy $\mu$ of the max-player is a sequence $(\mu_h)_{h \in [H]}$ of mappings from $\mathcal{X}_h$ to $\Delta_{\mathcal{A}}$. ($\mathcal{X}_h \subset \mathcal{X}$ is defined later.) A policy $\nu$ of the min-player is defined similarly. We let $\Pi_{\max}$ and $\Pi_{\min}$ denote the sets of max- and min-player's policies, respectively. The $t$-th episode proceeds as follows: an initial state $s_1^t$ is sampled from $p_0$. At the step $h$, the max- and min-player (resp.) observe their information sets $x_h^t := x(s_h^t)$ and $y_h^t := y(s_h^t)$. Given the information, the max- and min-player (resp.) choose and execute actions $a_h^t \sim \mu_h^t(\cdot|x_h)$ and $b_h^t \sim \nu_h^t(\cdot|y_h)$. As a result, the current state transitions to a next state $s_{h+1}^t \sim p_h(\cdot|s_h^t, a_h^t, b_h^t)$, and the max- and min-player receive rewards $r_h^t := r_h(s_h^t, a_h^t, b_h^t)$ and $-r_h^t$, respectively. This is repeated until a time step $H$, at which the episode finishes.

**Tree-like game structure and perfect recall assumption**   We assume that the game has a tree-like structure: for any state $s \in \mathcal{S}$, there is a unique step $h$ and history $(s_1, a_1, b_1, \ldots, s_h = s)$ to reach $s$. Precisely, for any policy of the players, for any realization of the game (i.e., trajectory) $(s_k', a_k', b_k')_{k \in [H]}$, conditionally to $s_i' = s$, it almost surely holds that $i = h$ and $(s_1', \ldots, s_h') = (s_1, \ldots, s_h)$. We also assume perfect recall, which means that each player remembers its past observations and actions. For example, in case of the max-player, for each information set $x \in \mathcal{X}$ there is a unique history $(x_1, a_1, \ldots, x_h = x)$ up to $x$. These assumptions require that $\mathcal{X}$ can be partitioned to $H$ subsets $(\mathcal{X}_h)_{h \in [H]}$ such that $x_h \in \mathcal{X}_h$ is reachable only at time step $h$; otherwise there would be two different histories up to $x_h$. $\mathcal{S}$ and $\mathcal{Y}$ can be also partitioned into $H$ subsets $(\mathcal{S}_h)_{h \in [H]}$, and $(\mathcal{Y}_h)_{h \in [H]}$, respectively.

Given the assumptions above, there exists a unique history $(s_1, a_1, b_1, \ldots, s_h = s, a_h = a, b_h = b)$ ending with $(s_h = s, a_h = a, b_h = b)$ for any state $s \in \mathcal{S}_h$, the max-player's action $a \in \mathcal{A}$, and the min-player's action $b \in \mathcal{B}$. Accordingly, the probability of $s_h = s, a_h = a, b_h = b$ can be computed by $p_h^{\mu,\nu}(s, a, b) = p_{1:h}(s)\mu_{1:h}(s, a)\nu_{1:h}(s, b)$, where

$$p_{1:h}(s) := p_0(s_1) \prod_{h'=1}^{h-1} p_{h'}(s_{h'+1}|s_{h'}, a_{h'}, b_{h'}),$$

$$\mu_{1:h}(s, a) := \mu_{1:h}(x(s), a) := \prod_{h'=1}^{h} \mu_{h'}(a_{h'} | x(s_{h'})),$$

$$\nu_{1:h}(s, b) := \nu_{1:h}(y(s), b) := \prod_{h'=1}^{h} \nu_{h'}(b_{h'} | y(s_{h'})).$$

With abuse of notation, we let $\mu_{1:h-1}(s) := \mu_{1:h-1}(x(s)) := \mu_{1:h-1}(s_{h-1}, a_{h-1})$, $p_h^{\mu,\nu}(s) := p_{1:h}(s)\mu_{1:h-1}(s, a)\nu_{1:h-1}(s, b)$ and $p_h^{\mu,\nu}(x) := \sum_{s \in x(s)} p_h^{\mu,\nu}(s)$ for any information set $x \in \mathcal{X}_h$. We use $\nu_{1:h-1}$ similarly.

**Bandit feedback**   We assume that the value of $r_h(s, a, b)$ is revealed to the players only when actions $a \in \mathcal{A}$ and $b \in \mathcal{B}$ are taken in a state $s \in \mathcal{S}$ at time step $h$. Notice that the players are

not aware of the underlying state. Furthermore, we assume that the players know neither the state transition dynamics nor the set of states $\mathcal{S}$. Such limitations impose a significant difficulty as the players need to carefully play the game trying different actions to gain the information of the game.

**Remark 1.** *Jin et al. (2020) consider a similar setting (from the view point of the max-player) of learning adversarial MDPs with bandit feedback wherein the reward function is chosen by an adversary. Our setting is different in that the players have only* imperfect information, *and that the* state transition dynamics is changing due to the learning opponents. *Nonetheless, the tree structure and perfect-recall assumptions allow a simple and efficient model-free algorithm that we provide.*

**Remark 2.** *Another recent line of works, Bai and Jin (2020); Bai et al. (2020); Liu et al. (2020), consider* perfect *information Markov game with bandits feedback, whereas our setting is the imperfect information one. By setting each information set of both players to be a singleton of the state, the perfect information setting is recovered. However, we assume perfect recall and the tree structure of the game. Although those assumptions are standard in game theory, they make the direct comparison of our setting and theirs impossible.*

**Regret and Nash Equilibrium (NE)**     For policies $\mu$ and $\nu$ we define the expected return (of the max-player) by $V^{\mu,\nu} := \sum_{h=1}^{H} \sum_{s_h \in \mathcal{S}_h, a_h \in \mathcal{A}, b_h \in \mathcal{B}} p_h^{\mu,\nu}(s_h, a_h, b_h) r_h(s_h, a_h, b_h)$. For sequences of policies $(\mu^t)_{t \in [T]} \in \Pi_{\max}^T$ and $(\nu^t)_{t \in [T]} \in \Pi_{\min}^T$, the regret of the max-player, relative to some policy $\mu^\dagger \in \Pi_{\max}$, is defined as

$$\mathfrak{R}_{\max}^T(\mu^\dagger) := \sum_{t=1}^{T} \left( V^{\mu^\dagger, \nu^t} - V^{\mu^t, \nu^t} \right). \tag{1}$$

Similarly, $\sum_{t=1}^{T} (V^{\mu^t, \nu^t} - V^{\mu^t, \nu^\dagger})$ is the min-player's regret relative to some $\nu^\dagger \in \Pi_{\min}$.

Our aim is to compute a NE. The following well-known folklore theorem,[6] which we prove in Appendix A, states that this problem can be converted into a regret minimization problem.

**Theorem 1.** *For each $h \in [H]$, $(x_h, a_h) \in \mathcal{X}_h \times \mathcal{A}$, and $(y_h, b_h) \in \mathcal{Y}_h \times \mathcal{B}$, define the average profile $(\overline{\mu}, \overline{\nu})$ by*

$$\overline{\mu}_h(a_h|x_h) := \frac{\sum_{t=1}^{T} \mu_{1:h}^t(x_h, a_h)}{\sum_{t=1}^{T} \mu_{1:h-1}^t(x_h)} \quad and \quad \overline{\nu}_h(b_h|y_h) := \frac{\sum_{t=1}^{T} \nu_{1:h}^t(y_h, b_h)}{\sum_{t=1}^{T} \nu_{1:h-1}^t(y_h)}, \tag{2}$$

*if the sum of the denominator is non-zero, otherwise as the uniform distribution over actions. If for some non-negative real value $\varepsilon$, we have that $(\mathfrak{R}_{\max}^T(\mu^\dagger) + \mathfrak{R}_{\min}^T(\nu^\dagger))/T \le \varepsilon$ for any profile $(\mu^\dagger, \nu^\dagger)$, then $(\overline{\mu}, \overline{\nu})$ are an $\varepsilon$-NE, i.e., $\max_{\mu \in \Pi_{\max}} V^{\mu,\overline{\nu}} - \min_{\nu \in \Pi_{\min}} V^{\overline{\mu},\nu} \le \varepsilon$.*

Given Theorem 1, we consider how to minimize the regret for the max- and min-player; or how to control the regret such that it grows sublinearly. The subsequent section presents an algorithm, which we call implicit exploration online mirror descent (`IXOMD`), that accomplishes this goal.

# 3   Implicit Exploration Online Mirror Descent (`IXOMD`)

Due to the symmetry of the players, it suffices to consider only the learning of the max-player. Therefore, we mainly focus on it and denote the max-player's regret (1) by $\mathfrak{R}^T(\mu^\dagger)$. We first convert the original regret minimization problem into a adversarial linear bandit one. Then, we give an explanation behind the use of implicit exploration and introduce our algorithm, `IXOMD`, whose pseudocode is given in Algorithm 1. For simplicity, we first give a simple-to-read but inefficient version. In Appendix F, we provide a practical version, whose computational and memory complexity are detailed in Section 3.3.

**Additional notation**     For a policy $\mu \in \Pi_{\max}$ and a sequence of functions $f := (f_h)_{h \in [H]}$, where $f_h : \mathcal{X}_h \times \mathcal{A} \to \mathbb{R}$, we denote the scalar product $\sum_{h \in [H]} \sum_{x_h \in \mathcal{X}_h, a \in \mathcal{A}} \mu_{1:h}(x_h, a) f_h(x_h, a)$ by $\langle \mu, f \rangle$. We let $\mathcal{F}^{t-1}$ be the $\sigma$-algebra generated by variables up to the beginning of the $t$-th episode, i.e., $\{s_h^\tau, a_h^\tau, b_h^\tau\}_{h \in [H], \tau \in [t-1]}$. We let $\mathbb{E}^{t-1}[\cdot] := \mathbb{E}[\cdot|\mathcal{F}^{t-1}]$.

---

[6]For example, see Farina et al. (2019) or Lanctot et al. (2009).

**Algorithm 1:** `IXOMD` for the Max-Player

**Input:** IX hyper-parameter $\gamma \in (0, \infty)$ and `OMD`'s learning rate $\eta \in (0, \infty)$.
**Output:** A near-NE policy for the max-player.

1 Initialize $\mu_h^1(a_h|x_h) \leftarrow 1/A$ for each $(x_h, a_h, h) \in \mathcal{X}_h \times \mathcal{A} \times [H]$.
2 **for** $t = 1, \ldots, T$ **do**
3     **for** $h = 1, \ldots, H$ **do**
4         Observe $x_h^t$, execute $a_h^t \sim \mu_h^t(\cdot|x_h^t)$, and receive $r_h^t$.
5     **end**
6     Set $Z_{H+1}^t \leftarrow 1$.
7     **for** $h = H, \ldots, 1$ **do**
8         Construct the IX loss estimate $\widetilde{\ell}_h^t$ by

$$\widetilde{\ell}_h^t \leftarrow \frac{1 - r_h^t}{\mu_{1:h}^t(x_h^t, a_h^t) + \gamma}.$$

9         For each $h \in [H]$ (with $Z_{H+1}^t \leftarrow 1$)

$$Z_h^t \leftarrow 1 - \mu_h^t(a_h^t|x_h^t) + \mu_h^t(a_h^t|x_h^t) \exp\left(-\eta\widetilde{\ell}_h^t + \log Z_{h+1}^t\right).$$

10         Update $\mu^t$ to $\mu^{t+1}$ at $x_h^t$ by

$$\mu_h^{t+1}(a_h|x_h^t) \leftarrow \begin{cases} \mu_h^t(a_h|x_h^t) \exp\left(-\eta\widetilde{\ell}_h^t + \log Z_{h+1}^t - \log Z_h^t\right) & \text{if } a_h = a_h^t \\ \mu_h^t(a_h|x_h^t) \exp(-\log Z_h^t) & \text{otherwise} \end{cases}$$

11         and $\mu^{t+1}(\cdot|x_h) \leftarrow \mu^t(\cdot|x_h)$ at other information sets $x_h \in \mathcal{X}_h$.
12     **end**
13 **end**
14 **return** *Policy* $\overline{\mu}$ *which is the average of* $\mu_1, \ldots, \mu_T$ *defined in Theorem* 1.

## 3.1 Conversion to online linear regret minimization

Note that for any profile $(\mu, \nu)$, we have

$$V^{\mu,\nu} = \sum_{h=1}^{H} \sum_{s_h \in \mathcal{S}_h, a_h \in \mathcal{A}, b_h \in \mathcal{B}} p_{1:h}(s_h)\mu_{1:h}(s_h, a_h)\nu_{1:h}(s_h, b_h)r_h(s_h, a_h, b_h)$$

$$= \sum_{h=1}^{H} \sum_{x_h \in \mathcal{X}_h, a_h \in \mathcal{A}} \mu_{1:h}(x_h, a_h) \sum_{s_h \in x_h, b_h \in \mathcal{B}} p_{1:h}(s_h)\nu_{1:h}(s_h, b_h)r_h(s_h, a_h, b_h),$$

where we used the facts that $\mu_{1:h}$ is dependent on $(x_h, a_h)$ rather than $(s_h, a_h)$, and $\sum_{s_h \in \mathcal{S}_h} f(s_h) = \sum_{x_h \in \mathcal{X}_h} \sum_{s_h \in x_h} f(s_h)$ for any function $f : \mathcal{S} \to \mathbb{R}$. Therefore defining a loss by

$$\ell_h^t(x_h, a_h) := \sum_{s_h \in x_h, b_h \in \mathcal{B}} p_{1:h}(s_h)\nu_{1:h}^t(s_h, b_h)(1 - r_h(s_h, a_h, b_h)),$$

we can rewrite the regret (1) as[7]

$$\mathfrak{R}^T(\mu^\dagger) = \sum_{t=1}^{T} \left\langle \mu^t - \mu^\dagger, \ell^t \right\rangle. \tag{3}$$

---

[7]As introduced at **Additional notation**, $\langle \mu^t, \widetilde{\ell}^t \rangle = \sum_{h=1}^{H} \sum_{x_h \in \mathcal{X}_h, a \in \mathcal{A}} \mu_{1:h}^t(x_h, a_h)\widetilde{\ell}_h^t(x_h, a_h)$. Hence the meaning of $\mu^t$ here is abused, and we are viewing it as a sequence $(\mu_{1:h}^t)_{h \in [H]}$ of functions. In this case, $\mu^t$ must satisfy the following two conditions: (non-negativity) $\mu_{1:h}^t(x_h, a_h) \geq 0$ for any $x_h \in \mathcal{X}_h$ and $h \in [H]$; (consistency) $\sum_{a_h \in \mathcal{A}} \mu_{1:h}^t(x_h, a_h) = \mu_{1:h-1}^t(x_{h-1}, a_{h-1})$ for any $x_h \in \mathcal{X}_h$ and $h \in \{2, \ldots, H\}$, where $(x_{h-1}, a_{h-1})$ is a unique predecessor of $x_h$, and $\sum_{a_1 \in \mathcal{A}} \mu_{1:1}^t(x_1, a_1) = 1$ for any $x_1 \in \mathcal{X}_1$. Nonetheless there is a bijective mapping between $\Pi_{\max}$ and the set of $\mu^t$ satisfying these two conditions. Therefore we do not discern these two sets.

This result tells us that we may convert the original regret minimization problem to a linear one in which we choose $\mu^t$ such that $\mathfrak{R}^T(\mu^\dagger)$ grows sublinearly. Note that, as mentioned in the introduction, this reduction is not new and can be traced back to the work by Romanovsky (1962); von Stengel (1996). It is important to remark that the losses are bounded in the unit interval. See Appendix E for a proof of the following lemma.

**Lemma 2.** *For all $t, h, x_h, a_h$ the loss is bounded $\ell_h^t(x_h, a_h) \in [0, 1]$.*

### 3.2 Loss estimation and implicit exploration

To solve the regret minimization problem (3) with bandit feedback, we need to estimate $\ell^t$. An unbiased importance sampling estimator is

$$\widehat{\ell}_h^t(x_h, a_h) := \frac{\mathbb{I}_{\{x_h = x_h^t, a_h = a_h^t\}}}{\mu_{1:h}^t(x_h, a_h)}\left(1 - r_h^t\right). \tag{4}$$

However, instead, we estimate the loss by

$$\widetilde{\ell}_h^t(x_h, a_h) := \frac{\mathbb{I}_{\{x_h = x_h^t, a_h = a_h^t\}}}{\mu_{1:h}^t(x_h, a_h) + \gamma}\left(1 - r_h^t\right), \tag{5}$$

where $\gamma$ is a positive real value and a hyper-parameter. This estimator is used by implicit exploration in bandits (IX, Kocák et al., 2014; Neu, 2015; Lattimore and Szepesvári, 2020, Chapter 12), and we therefore refer to it as the IX estimator. Note that IX uses a biased estimate, but it prevents the variance of the IX estimator from becoming too large.

### 3.3 Efficient implementation, Space- and Time-Complexities

Given a loss estimate, we find $\mu^{t+1}$ by solving

$$\mu^{t+1} := \underset{\mu \in \Pi_{\max}}{\arg\min}\, \eta\left\langle \mu, \widetilde{\ell}^t \right\rangle + \mathrm{D}\left(\mu \| \mu^t\right), \tag{6}$$

where $\mathrm{D}$ is the *dilated* entropy distance-generating function (with uniform weights, Kroer et al. (2015)) defined by

$$\mathrm{D}(\mu \| \mu') := \sum_{h=1}^{H} \sum_{x_h \in \mathcal{X}_h, a_h \in \mathcal{A}} \mu_{1:h}(x_h, a_h) \log\frac{\mu_h(a_h|x_h)}{\mu_h'(a_h|x_h)}\,.$$

Note that $\mathrm{D}$ is a Bregman divergence, see Lemma 9 in Appendix E. The update in (6) has an easy implementation, as explained next. For more details of its derivation, please refer to Appendix C. To compute a new policy, we first need to compute for each $h \in [H]$,

$$Z_h^t := \sum_{a_h \in \mathcal{A}} \mu_h^t(a_h|x_h^t)\exp\left(\mathbb{I}_{\{a_h = a_h^t\}}\left(-\eta\widetilde{\ell}_H^t(x_h^t, a_h) + \log Z_{h+1}^t\right)\right)$$

$$= 1 - \mu_h^t(a_h^t|x_h^t) + \mu_h^t(a_h^t|x_h^t)\exp\left(-\eta\widetilde{\ell}_H^t(x_h^t, a_h^t) + \log Z_{h+1}^t\right), \tag{7}$$

with $Z_{H+1}^t := 1$. Then, we can compute a new policy by

$$\mu_h^{t+1}(a_h|x_h^t) = \mu_h^t(a_h|x_h^t)\exp\left(\mathbb{I}_{\{a_h = a_h^t\}}\left(-\eta\widetilde{\ell}_h^t(x_h^t, a_h) + \log Z_{h+1}^t\right) - \log Z_h^t\right). \tag{8}$$

Note that this policy is updated *only* at the information sets visited along the $t$-th trajectory. This implies that the update requires $\mathcal{O}(HA)$ time-complexity per episode. Therefore the learning of the policies require $\mathcal{O}(THA)$ time-complexity in total.

Interestingly, the update of the average policy $\overline{\mu}$ can also be performed in a semi-online way, see Appendix D. This method has a total time-complexity of $\mathcal{O}(THA + \min(TH, X)A)$ and space-complexity of $\mathcal{O}(\min(TH, X)A)$. Please refer to Algorithm 3 in Appendix F for a pseudocode of this practical implementation.

Algorithm 3 requires a post-hoc computation that is the source of $\mathcal{O}(\min(TH, X)A)$ time-complexity. It is possible to defer the post-hoc computation until $\overline{\mu}(\cdot|x_h)$ is needed for playing a game. In this case, the computation of $\overline{\mu}(\cdot|x_h)$ is performed while traversing a game tree. For one traversal, $\overline{\mu}(\cdot|x_h)$ is computed for each $h$, and the total time-complexity is $\mathcal{O}(HA)$. The space-complexity is unchanged and is $\mathcal{O}(\min(TH, X)A)$.

# 4 Theoretical Analysis of `IXOMD`

We now analyze `IXOMD`. It has the following guarantee, which we shall prove in the present section.

**Theorem 3** (regret bound of `IXOMD`). *Let $\delta \in (0,1)$. The regret (1) satisfies the following guarantee with probability at least $1 - \delta$*

$$\max_{\mu^\dagger \in \Pi_{\max}} \mathfrak{R}^T(\mu^\dagger) \leq H\sqrt{2T\iota} + \gamma TXA + \frac{X\iota}{2\gamma} + \frac{X\log A}{\eta} + \eta HTXA + \frac{\eta H^2 \iota}{2\gamma},$$

*where $\iota := \log(3XA/\delta)$. In particular $\eta = \sqrt{\dfrac{\log A}{THA}}$ and $\gamma = \sqrt{\dfrac{\iota}{2TA}}$ result in*

$$\max_{\mu^\dagger \in \Pi_{\max}} \mathfrak{R}^T(\mu^\dagger) \leq H\sqrt{2T\iota} + X\sqrt{2TA\iota} + X\sqrt{THA\log A} + H\sqrt{\frac{H\iota \log A}{2}}.$$

**Remark 3.** *We emphasize that this result is agnostic of the min-player. In particular, the same result holds for learning in a partially observable MDP with adversarial state-transition dynamics and reward function, as long as assumptions similar to the tree-like structure and perfect recall hold.*

**Remark 4.** *In Theorem 3, we adjusted $\eta$ and $\gamma$ using $T$, $H$, $X$, and $A$. Even when we know $T$ only, setting $\eta = 1/\sqrt{T}$ and $\gamma = 1/\sqrt{T}$ guarantees an upper-bound of the order of $\widetilde{\mathcal{O}}(XA\sqrt{T})$[8]. If we additionally know $H$ and $A$ (which is likely to be the case), but do not know $X$, setting $\eta = \sqrt{\log A/(THA)}$ and $\gamma = 1/\sqrt{2TA}$ still results in an upper-bound of the order of $\widetilde{\mathcal{O}}(X\sqrt{TA})$.*

A similar result holds for the min-player thanks to the symmetry. From Theorem 1 and 3, it follows that the average profile $(\overline{\mu}, \overline{\nu})$ is close to a Nash equilibrium with high probability.

**Corollary 3.1.** *Suppose that both max- and min-players learn their policies by `IXOMD` with the setting[9] of $\eta$ and $\gamma$ in Theorem 3. Then with probability at least $1 - \delta$, the average profile $(\overline{\mu}, \overline{\nu})$ defined in Theorem 1 is $\varepsilon$-Nash equilibrium, with*

$$\varepsilon := \widetilde{\mathcal{O}}\left(\frac{1}{\sqrt{T}}\left(X\sqrt{A} + Y\sqrt{B}\right)\right).$$

## 4.1 Proof of Theorem 3

Now we start the proof of Theorem 3. In the first step, we decompose the regret (3) to three terms:

$$\mathfrak{R}^T(\mu^\dagger) = \underbrace{\sum_{t=1}^T \left\langle \mu^t, \ell^t - \widetilde{\ell}^t \right\rangle}_{\text{BIAS 1}} - \underbrace{\sum_{t=1}^T \left\langle \mu^\dagger, \ell^t - \widetilde{\ell}^t \right\rangle}_{\text{BIAS 2}} + \underbrace{\sum_{t=1}^T \left\langle \mu^t - \mu^\dagger, \widetilde{\ell}^t \right\rangle}_{\text{REGRET}}. \tag{9}$$

Then, we prove a high-probability upper-bound for each term. After deriving each upper-bound, Theorem 3 follows simply by taking the union bound over the three terms.

For proving the upper-bounds, we need the following lemma, which almost immediately follows from Lemma 1 by Neu (2015) (also see Lemma 12.2 of Lattimore and Szepesvári (2020) for a more general statement). For completeness we prove it in Appendix E.

**Lemma 4.** *Let $\delta \in (0,1)$ and $\gamma \in (0, \infty)$. Fix $h \in [H]$, and let $\alpha^t(x_h, a_h) \in [0, 2\gamma]$ be $\mathcal{F}^{t-1}$-measurable random variable for each $(x_h, a_h) \in \mathcal{X}_h \times \mathcal{A}$. Then with probability at least $1 - \delta$*

$$\sum_{t=1}^T \sum_{x_h \in \mathcal{X}_h, a_h \in \mathcal{A}} \alpha^t(x_h, a_h)\left(\widetilde{\ell}_h^t(x_h, a_h) - \ell_h^t(x_h, a_h)\right) \leq \log\frac{1}{\delta}$$

We first prove an upper-bound of BIAS 1 shown below.

---

[8]We recall that we hide with $\widetilde{\mathcal{O}}$ poly-log terms in $e^H, T, X, A, 1/\delta$.

[9]Note that $A$ and $X$ must be replaced with $B$ and $Y$ (resp.) for the min-player's $\eta$ and $\gamma$. Also note that to archive the same order of a bound as the one shown in this corollary, we need neither $X$ nor $Y$.

**Lemma 5** (upper-bound of BIAS 1). *Let $\delta \in (0,1)$. It holds with probability at least $1 - \delta/3$ that BIAS 1 $\leq H\sqrt{2T\iota} + \gamma TXA$.*

*Proof.* To see that this is true, we first deduce that

$$
\left\langle \mu^t, \widetilde{\ell}^t \right\rangle = \sum_{h=1}^{H} \sum_{x_h \in \mathcal{X}_h, a_h \in \mathcal{A}} \mu_{1:h}^t(x_h, a_h) \frac{\mathbb{I}_{\{x_h = x_h^t, a_h = a_h^t\}}}{\mu_{1:h}^t(x_h, a_h) + \gamma} \left(1 - r_h^t\right)
$$

$$
\leq \sum_{h=1}^{H} \sum_{x_h \in \mathcal{X}_h, a_h \in \mathcal{A}} \mathbb{I}_{\{x_h = x_h^t, a_h = a_h^t\}} = \sum_{h=1}^{H} 1 = H \,,
$$

where the inequality follows from facts that $\mu_{1:h}^t(x_h, a_h)/(\mu_{1:h}^t(x_h, a_h) + \gamma) \leq 1$, and $0 \leq 1 - r_h^t \leq 1$. By Hoeffding-Azuma inequality, we deduce that $\sum_{t=1}^{T} \langle \mu^t, \widetilde{\ell}^t - \mathbb{E}^{t-1}[\widetilde{\ell}^t] \rangle \geq -H\sqrt{2T \log(3/\delta)} \geq -H\sqrt{2T\iota}$ with probability at least $1 - \delta/3$. (The final inequality is to simplify the result.) Next, we deduce that

$$
\left\langle \mu^t, \ell^t - \mathbb{E}^{t-1}\!\left[\widetilde{\ell}^t\right] \right\rangle = \sum_{h=1}^{H} \sum_{x_h \in \mathcal{X}_h, a_h \in \mathcal{A}} \mu_{1:h}^t(x_h, a_h)\left(1 - \frac{\mu_{1:h}^t(x_h, a_h)}{\mu_{1:h}^t(x_h, a_h) + \gamma}\right) \ell_h^t(x_h, a_h)
$$

$$
= \sum_{h=1}^{H} \sum_{x_h \in \mathcal{X}_h, a_h \in \mathcal{A}} \mu_{1:h}^t(x_h, a_h) \frac{\gamma \ell_h^t(x_h, a_h)}{\mu_{1:h}^t(x_h, a_h) + \gamma}
$$

$$
\leq \gamma \sum_{h=1}^{H} \sum_{x_h \in \mathcal{X}_h, a_h \in \mathcal{A}} \ell_h^t(x_h, a_h) \leq \gamma \sum_{h=1}^{H} \sum_{x_h \in \mathcal{X}_h, a_h \in \mathcal{A}} 1 \leq \gamma XA \,,
$$

where the first inequality follows from $\mu^t(x_h, a_h)/(\mu_{1:h}^t(x_h, a_h) + \gamma) \leq 1$, and the last inequality follows from $\sum_{h=1}^{H} \sum_{x_h \in \mathcal{X}_h, a_h \in \mathcal{A}} 1 = \sum_{h=1}^{H} |\mathcal{X}_h| A = XA$. Combining both bounds, we obtain the claimed result. □

Next we prove an upper-bound of BIAS 2.

**Lemma 6** (upper-bound of BIAS 2). *Let $\delta \in (0,1)$. For any $\mu^\dagger \in \Pi_{\max}$ it holds with probability at least $1 - \delta/3$ that BIAS 2 $\leq X\iota/(2\gamma)$.*

*Proof.* Note that

$$
\sum_{t=1}^{T} \sum_{h=1}^{H} \sum_{x_h \in \mathcal{X}_h, a \in \mathcal{A}} \mu_{1:h}^\dagger(x_h, a_h)\left(\widetilde{\ell}_h^t(x_h, a_h) - \ell_h^t(x_h, a_h)\right)
$$

$$
= \sum_{h=1}^{H} \sum_{x_h \in \mathcal{X}_h, a_h \in \mathcal{A}} \mu_{1:h}^\dagger(x_h, a_h) \underbrace{\sum_{t=1}^{T} \sum_{x_h' \in \mathcal{X}_h, a_h' \in \mathcal{A}} \mathbb{I}_{\{x_h' = x_h, a_h' = a_h\}}\left(\widetilde{\ell}_h^t(x_h', a_h') - \ell_h^t(x_h', a_h')\right)}_{\clubsuit} \,.
$$

Now we can apply Lemma 4 to $\clubsuit$ and deduce that, for each $(x_h, a_h)$, we have

$$
\sum_{t=1}^{T} \sum_{x_h' \in \mathcal{X}_h, a_h' \in \mathcal{A}} \mathbb{I}_{\{x_h' = x_h, a_h' = a_h\}}\left(\widetilde{\ell}_h^t(x_h', a_h') - \ell_h^t(x_h', a_h')\right) \leq \frac{\iota}{2\gamma}
$$

with probability at least $1 - \delta/(3XA)$. We deduce that

$$
\sum_{t=1}^{T} \sum_{h=1}^{H} \sum_{x_h \in \mathcal{X}_h, a \in \mathcal{A}} \mu_{1:h}^\dagger(x_h, a_h)\left(\widetilde{\ell}_h^t(x_h, a_h) - \ell_h^t(x_h, a_h)\right)
$$

$$
\leq \frac{\iota}{2\gamma} \sum_{h=1}^{H} \sum_{x_h \in \mathcal{X}_h, a_h \in \mathcal{A}} \mu_{1:h}^\dagger(x_h, a_h) \leq \frac{X\iota}{2\gamma}
$$

with probability at least $1 - \delta/3$, using a union bound over all $(x_h, a_h) \in X_h \times A$ and each $h$. □

Finally we prove the following upper-bound of REGRET in Appendix B.

**Lemma 7** (upper-bound of REGRET). *Let $\delta \in (0, 1)$. For any $\mu^\dagger \in \Pi_{\max}$ it holds with probability at least $1 - \delta/3$ that*

$$REGRET \leq \frac{X \log A}{\eta} + \eta H T X A + \frac{\eta H^2 \iota}{2\gamma} \,.$$

## 5 Conclusion

We theoretically studied the problem of learning a NE of an IIG under a perfect-recall assumption. We provided the `IXOMD` algorithm based on `OMD` with the dilated entropy distance-generating function as a regularizer and implicit exploration for estimation of the losses. We proved a high-probability bound on the convergence rate to the NE of order $\widetilde{\mathcal{O}}(X\sqrt{A} + Y\sqrt{B})/\sqrt{T}$ derived from a regret bound of order $\widetilde{\mathcal{O}}(X\sqrt{AT})$ (for the max-player). Notably, the regret bound remains valid in the adversarial setting (where the opponent and the game are picked by an adversary). Furthermore, due to our choice of the regularizer, the updates of the policy (e.g., of the max-player) could be implemented with a time-complexity of $\mathcal{O}(HA)$ per episode, which makes `IXOMD` also computationally efficient. Precisely, the total time complexity (after $T$ episodes) is of order $\mathcal{O}(TH(A+B)+\min(TH, X)A+\min(TH, Y)B)$ while the space complexity is of order $\mathcal{O}(\min(TH, X)A + \min(TH, Y)B)$.

An interesting next direction of research would be to characterize the problem-independent optimal regret, e.g., for the max-player, in our setting. We conjecture that it is of order $\widetilde{\mathcal{O}}(\sqrt{XAT})$ even in the adversarial setting (where the opponent and the game are picked by an adversary). This would make our current bound to be loose by a factor $\sqrt{X}$.

## Acknowledgments

Tadashi Kozuno gratefully acknowledges funding from the Canada CIFAR AI Chairs Program, Amii, and NSERC. Pierre Ménard is supported by the SFI Sachsen-Anhalt for the project REBCI ZS/2019/10/102024 by the Investitionsbank SachsenAnhalt.

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
