# A  Proof of the Folklore Theorem 1

For completeness, in this appendix we provide a proof of Theorem 1, which is a well-known folklore theorem. Notice that $V^{\mu,\nu}$ is *linear* in a policy $\mu \in \Pi_{\max}$, that is, $V^{\mu,\nu} = \langle \mu, r^\nu \rangle$, where we define $r_h^\nu(x_h, a_h) := \sum_{s_h \in x_h, b_h \in \mathcal{B}} p_{1:h}(s_h) \nu_{1:h}(s_h, b_h) r_h(s_h, a_h, b_h)$. By definition, the regret of the min-player relative to some policy $\nu^\dagger \in \Pi_{\min}$ is given as

$$\mathfrak{R}_{\min}^T(\nu^\dagger) = \sum_{t=1}^T \left( \left\langle \mu^t, r^{\nu^t} \right\rangle - \left\langle \mu^t, r^{\nu^\dagger} \right\rangle \right) = \sum_{t=1}^T \left\langle \mu^t, r^{\nu^t} \right\rangle - T \left\langle \frac{1}{T} \sum_{t=1}^T \mu^t, r^{\nu^\dagger} \right\rangle.$$

Define $\overline{\mu}_{1:h}(x_h, a_h) := \overline{\mu}_{1:h-1}(x_h)\overline{\mu}_h(a_h|x_h) := \prod_{h'=1}^h \overline{\mu}_{h'}(a_{h'}|x_{h'})$, where $(x_{h'}, a_{h'})_{h' \in [h-1]}$ is a unique history up to $(x_h, a_h) \in \mathcal{X}_h \times \mathcal{A}$, similarly to $\mu_{1:h}$ for a policy $\pi \in \Pi_{\max}$. The above expression of the min-player's regret tells us that if

$$\overline{\mu}_{1:h}(x_h, a_h) = \frac{1}{T} \sum_{t=1}^T \mu_{1:h}^t(x_h, a_h) \tag{10}$$

holds for any $h \in [H]$ and any observation-action pair $(x_h, a_h) \in \mathcal{X}_h \times \mathcal{A}$, then

$$\mathfrak{R}_{\min}^T(\nu^\dagger) = \sum_{t=1}^T \left\langle \mu^t, r^{\nu^t} \right\rangle - T \left\langle \overline{\mu}, r^{\nu^\dagger} \right\rangle = \sum_{t=1}^T \left( V^{\mu^t, \nu^t} - V^{\overline{\mu}, \nu^\dagger} \right).$$

A similar result holds for the regret of the max-player, and we have

$$\max_{\mu^\dagger \in \Pi_{\max}} V^{\mu^\dagger, \overline{\nu}} - \min_{\nu^\dagger \in \Pi_{\min}} V^{\overline{\mu}, \nu^\dagger}$$

$$= \max_{\mu^\dagger \in \Pi_{\max}} \frac{1}{T} \sum_{t=1}^T \left( V^{\mu^\dagger, \nu^t} - V^{\mu^t, \nu^t} \right) - \min_{\nu^\dagger \in \Pi_{\min}} \frac{1}{T} \sum_{t=1}^T \left( V^{\mu^t, \nu^\dagger} - V^{\mu^t, \nu^t} \right)$$

$$= \frac{1}{T} \left( \max_{\mu^\dagger \in \Pi_{\max}} \mathfrak{R}_{\max}^T(\mu^\dagger) + \max_{\nu^\dagger \in \Pi_{\min}} \mathfrak{R}_{\min}^T(\nu^\dagger) \right) \le \varepsilon.$$

Therefore, $(\overline{\mu}, \overline{\nu})$ is an $\varepsilon$-NE.

We now prove Equation 10 by induction over $h$. This property is obviously true for $h = 1$ from the definition of the average profile (2). Now assume Equation 10 holds for any observation-action pair $(x_{h'}, a_{h'})$ of depth $h' < h$. Consider an observation $x_h \in \mathcal{X}_h$ of depth $h$. Let $(x_{h-1}, a_{h-1})$ be its immediate predecessor. Then, from the definition of $\overline{\mu}$ we have, for any $a_h \in \mathcal{A}$,

$$\overline{\mu}_{1:h}(x_h, a_h) = \overline{\mu}_{1:h-1}(x_h)\overline{\mu}_h(a_h|x_h)$$

$$= \overline{\mu}_{1:h-1}(x_{h-1}, a_{h-1}) \frac{\sum_{t=1}^T \mu_{1:h}^t(x_h, a_h)}{\sum_{t=1}^T \mu_{1:h-1}^t(x_h)}$$

$$= \frac{1}{T} \sum_{t=1}^T \mu_{1:h-1}^t(x_{h-1}, a_{h-1}) \frac{\sum_{t=1}^T \mu_{1:h}^t(x_h, a_h)}{\sum_{t=1}^T \mu_{1:h-1}^t(x_h)}$$

$$= \frac{1}{T} \sum_{t=1}^T \mu_{1:h-1}^t(x_h) \frac{\sum_{t=1}^T \mu_{1:h}^t(x_h, a_h)}{\sum_{t=1}^T \mu_{1:h-1}^t(x_h)}$$

$$= \frac{1}{T} \sum_{t=1}^T \mu_{1:h}^t(x_h, a_h).$$

Therefore, Equation 10 holds for any $h \in [H]$ and this concludes the proof of Theorem 1.

# B  Proof of Lemma 7

To prove the upper-bound, we first connect $\langle \mu^t - \mu^\dagger, \widetilde{\ell}_h^t \rangle$ to divergences between $\mu^\dagger$, $\mu^t$ and $\mu^{t+1}$. To this end the following technical lemma turns out to be useful.

**Lemma 8.** *For any policy $\mu \in \Pi_{\max}$ we have that*

$$\mathrm{D}\big(\mu\|\mu^{t+1}\big) - \mathrm{D}\big(\mu\|\mu^t\big) = \eta\Big\langle \mu, \widetilde{\ell}^t \Big\rangle + \log Z_1^t$$

*Proof.* From the form of policy updates (8) we may deduce that

$$\mathrm{D}\big(\mu\|\mu^{t+1}\big) - \mathrm{D}\big(\mu\|\mu^t\big)$$

$$= \sum_{h=1}^{H} \sum_{x_h \in \mathcal{X}_h, a_h \in \mathcal{A}} \mu_{1:h}(x_h, a_h) \log \frac{\mu_h^t(a_h|x_h)}{\mu_h^{t+1}(a_h|x_h)}$$

$$= \sum_{h=1}^{H} \mu_{1:h}(x_h^t, a_h^t)\Big(\eta\widetilde{\ell}_h^t(x_h^t, a_h^t) - \log Z_{h+1}^t\Big) + \sum_{h=1}^{H} \mu_{1:h-1}(x_h^t) \log Z_h^t .$$

By noting that

$$-\sum_{h=1}^{H} \mu_{1:h}(x_h^t, a_h^t) \log Z_{h+1}^t + \sum_{h=1}^{H} \mu_{1:h-1}(x_h^t) \log Z_h^t$$

$$= -\sum_{h=1}^{H-1} \mu_{1:h}(x_{h+1}^t) \log Z_{h+1}^t + \sum_{h=1}^{H} \mu_{1:h-1}(x_h^t) \log Z_h^t$$

$$= -\sum_{h=2}^{H} \mu_{1:h-1}(x_h^t) \log Z_h^t + \sum_{h=1}^{H} \mu_{1:h-1}(x_h^t) \log Z_h^t = \log Z_1^t ,$$

we deduce the claimed result. $\qquad\square$

Now we are ready to prove Lemma 7.

*proof of Lemma 7.* From a fact that

$$\mathrm{D}\big(\mu^\dagger\|\mu^t\big) - \mathrm{D}\big(\mu^\dagger\|\mu^{t+1}\big) + \mathrm{D}\big(\mu^t\|\mu^{t+1}\big)$$

$$= -\big(\mathrm{D}\big(\mu^\dagger\|\mu^{t+1}\big) - \mathrm{D}\big(\mu^\dagger\|\mu^t\big)\big) + \mathrm{D}\big(\mu^t\|\mu^{t+1}\big) - \mathrm{D}\big(\mu^t\|\mu^t\big) ,$$

and Lemma 8, we have that $\eta\langle \mu^t - \mu^\dagger, \widetilde{\ell}^t \rangle = \mathrm{D}\big(\mu^\dagger\|\mu^t\big) - \mathrm{D}\big(\mu^\dagger\|\mu^{t+1}\big) + \mathrm{D}\big(\mu^t\|\mu^{t+1}\big)$. Taking the sum over $t$ noting that $\mathrm{D}\big(\mu^\dagger\|\mu^{T+1}\big) \geq 0$, we deduce that

$$\eta \sum_{t=1}^{T} \Big\langle \mu^t - \mu^\dagger, \widetilde{\ell}^t \Big\rangle \leq \mathrm{D}\big(\mu^\dagger\|\mu^1\big) + \sum_{t=1}^{T} \mathrm{D}\big(\mu^t\|\mu^{t+1}\big) .$$

We need to upper-bound the two terms on the right side.

The first term is easy to upper-bound. From the definition of the divergence and the choice for the first policy we have

$$\mathrm{D}\big(\mu^\dagger\|\mu^1\big) = \sum_{h=1}^{H} \sum_{x_h \in \mathcal{X}_h, a_h \in \mathcal{A}} \mu_{1:h}^\dagger(x_h, a_h) \log \frac{\mu_h^\dagger(a_h|x_h)}{\mu_h^1(a_h|x_h)}$$

$$\leq -\sum_{h=1}^{H} \sum_{x_h \in \mathcal{X}_h, a_h \in \mathcal{A}} \mu_{1:h}^\dagger(x_h, a_h) \log \mu_h^1(x_h, a_h)$$

$$= \log A \sum_{h=1}^{H} \sum_{x_h \in \mathcal{X}_h, a_h \in \mathcal{A}} \mu_{1:h}^\dagger(x_h, a_h) \leq X \log A .$$

In contrast bounding the second term is somewhat lengthy and technical. For brevity we use the following notations: $\widetilde{\ell}_h^t := \widetilde{\ell}_h^t(x_h^t, a_h^t)$, $\mu_h^t := \mu_h^t(x_h^t, a_h^t)$ and $\mu_{h:h'}^t := \mu_{h'}^t/\mu_h^t$, where $h' > h$.

From Lemma 8 we have that

$$\mathrm{D}\big(\mu^t\|\mu^{t+1}\big) = \mathrm{D}\big(\mu^t\|\mu^{t+1}\big) - \mathrm{D}\big(\mu^t\|\mu^t\big) = \eta\Big\langle\mu^t,\widetilde{\ell}^t\Big\rangle + \log Z_1^t\,.$$

We show that $\log Z_1^t \approx -\eta\langle\mu^t,\widetilde{\ell}^t\rangle$. We introduce auxiliary independent Bernoulli random variables $z_h^t \sim \mathcal{B}\mathrm{er}(\mu_h^t)$ and the product $z_{h:h'} = \prod_{i\in[h,h']} z_i^t$. We can check that the following expectation is solution of the same recurrence relation (7) that defines $Z_h^t$,

$$\mathbb{E}_{(z_{h'}^t)_{h'\in[h,H]}}\left[\exp\!\left(-\eta\sum_{h'=h}^{H} z_{h:h'}^t\widetilde{\ell}_{h'}^t\right)\right] = (1-\mu_h^t) + \mu_h^t e^{-\eta\ell_h^t}\mathbb{E}_{(z_{h'}^t)_{h'\in[h+1,H]}}\left[\exp\!\left(-\eta\sum_{h'=h+1}^{H} z_{h+1:h'}^t\widetilde{\ell}_{h'}^t\right)\right].$$

Thus by recurrence we obtain for all $h\in[H]$

$$Z_h^t = \mathbb{E}_{(z_{h'}^t)_{h'\in[h,H]}}\left[\exp\!\left(-\eta\sum_{h'=h}^{H} z_{h:h'}^t\widetilde{\ell}_h^t\right)\right]\,.$$

Using successively, the previous equality for $h=1$, $\log(x)\le x-1$ and $e^{-x}\le 1-x+x^2$ for $x\ge 0$, Jensen inequality and the fact that $\mathbb{E}_{(z_h^t)_{h\in[1,H]}}[z_{1:h'}^t]=\mu_{h'}^t$ we get

$$\eta\Big\langle\mu^t,\widetilde{\ell}^t\Big\rangle + \log(Z_1^t) = \eta\Big\langle\mu^t,\widetilde{\ell}^t\Big\rangle + \log\mathbb{E}_{(z_h^t)_{h\in[1,H]}}\left[\exp\!\left(-\eta\sum_{h=1}^{H} z_{1:h}^t\widetilde{\ell}_h^t\right)\right]$$

$$\le \eta^2\mathbb{E}_{(z_h^t)_{h\in[1,H]}}\left[\left(\sum_{h=1}^{H} z_{1:h}^t\widetilde{\ell}_h^t\right)^2\right]$$

$$\le \eta^2 H\,\mathbb{E}_{(z_h^t)_{h\in[1,H]}}\left[\sum_{h=1}^{H} z_{1:h}^t\big(\widetilde{\ell}_h^t\big)^2\right]$$

$$= \eta^2 H\sum_{h=1}^{H}\mu_h^t\big(\widetilde{\ell}_h^t\big)^2\,.$$

Therefore, using $\mu_{1:h}^t\widetilde{\ell}_h^t\le 1$, it holds

$$\mathrm{D}\big(\mu^t\|\mu^{t+1}\big) \le \eta^2 H\sum_{h=1}^{H}\mu_h^t\big(\widetilde{\ell}_h^t\big)^2 \le \eta^2 H\sum_{h=1}^{H}\widetilde{\ell}_h^t\,.$$

Recalling that $\widetilde{\ell}_h^t$ is non-zero only at $(x_h^t,a_h^t)$, we have that $\widetilde{\ell}_h^t = \sum_{x_h\in\mathcal{X}_h,a_h\in\mathcal{A}}\widetilde{\ell}_h^t(x_h,a_h)$. Thus we can use Lemma 4, which implies

$$\eta^2 H\sum_{t=1}^{T}\sum_{h=1}^{H}\widetilde{\ell}_h^t \le \eta^2 H\sum_{t=1}^{T}\sum_{h=1}^{H}\sum_{x_h\in\mathcal{X}_h,a_h\in\mathcal{A}}\ell_h^t(x_h,a_h) + \frac{\eta^2 H^2\log(3H/\delta)}{2\gamma}$$

$$\le \eta^2 HTXA + \frac{\eta^2 H^2\iota}{2\gamma},$$

where at the final line we loosened the bound by replacing $\log(3H/\delta)$ with $\iota$ to simplify the bound. This concludes the proof.

$\square$

## C  Details of Efficient Implementation (Section 3.3)

In this appendix we prove that the update (6) corresponds to the policy update (8), which is shown here for convenience.

$$\mu_h^{t+1}(a_h|x_h^t) = \mu_h^t(a_h|x_h^t)\exp\!\Big(\mathbb{I}_{\{a_h=a_h^t\}}\big(-\eta\widetilde{\ell}_h^t(x_h^t,a_h) + \log Z_{h+1}^t\big) - \log Z_h^t\Big),$$

where

$$Z_h^t := \sum_{a_h \in \mathcal{A}} \mu_h^t(a_h|x_h^t) \exp\Big(\mathbb{I}_{a_h=a_h^t}\Big(-\eta\widetilde{\ell}_h^t(x_h^t,a_h) + \log Z_{h+1}^t\Big)\Big)$$

$$= 1 - \mu_h^t(a_h^t|x_h^t) + \mu_h^t(a_h^t|x_h^t) \exp\Big(-\eta\widetilde{\ell}_h^t(x_h^t,a_h^t) + \log Z_{h+1}^t\Big)$$

with $Z_{H+1}^t := 1$. Note that no policy updates occur at unvisited information sets.

We prove the correspondence by induction on $h$. Recall that $\widetilde{\ell}^t$ is non-zero only at visited information sets and actions $(x_h^t, a_h^t)_{h \in [H]}$. Therefore

$$\eta\big\langle \mu, \widetilde{\ell}^t \big\rangle + \mathrm{D}\big(\mu\|\mu^t\big) = \sum_{h=1}^{H}\Bigg(\eta\mu_{1:h}(x_h^t,a_h^t)\widetilde{\ell}_h^t(x_h^t,a_h^t) + \sum_{x_h \in \mathcal{X}_h} \mu_{1:h-1}(x_h)\,\mathrm{KL}\big(\mu_h\|\mu_h^t\big)(x_h)\Bigg),$$

where $\mathrm{KL}(\mu_h\|\mu_h^t)(x)$ is a shorthand notation for Kullback-Leibler divergence $\mathrm{KL}(\mu_h(\cdot|x)\|\mu_h^t(\cdot|x))$. Because it suffices to optimize $\mu$ at visited information sets, we may focus on terms involving them. Accordingly to find $\mu^{t+1}$ we need to minimize

$$\mathfrak{L}(\mu_1,\ldots,\mu_H) := \sum_{h=1}^{H} \mu_{1:h-1}(x_h^t)\Big(\eta\mu_h(a_h^t|x_h^t)\widetilde{\ell}_h^t(x_h^t,a_h^t) + \mathrm{KL}\big(\mu_h\|\mu_h^t\big)(x_h)\Big)$$

with respect to $\mu$. For $h = H$ it is straightforward to deduce that

$$\mu_H^{t+1}(a_H|x_H^t) = \mu_H^t(a_H|x_H^t) \exp\Big(-\eta\mathbb{I}_{\{a_H=a_H^t\}}\widetilde{\ell}_H^t(x_H^t,a_H) - \log Z_H^t\Big).$$

Assume that the claim holds up to step $h+1$. Then for $\mu$ such that $\mu_{h'} = \mu_{h'}^{t+1}$ for $h' > h$ we have

$$\mathfrak{L}(\mu_1,\ldots,\mu_H)$$

$$= \sum_{h'=1}^{h} \mu_{1:h'-1}(x_{h'}^t)\Big(\eta\mu_{h'}(a_{h'}^t|x_{h'}^t)\widetilde{\ell}_{h'}^t(x_{h'}^t,a_{h'}^t) + \mathrm{KL}\big(\mu_{h'}\|\mu_{h'}^t\big)(x_{h'}^t)\Big)$$

$$\quad + \sum_{h'=h+1}^{H} \mu_{1:h'-1}(x_{h'}^t)\Big(\eta\mu_{h'}(a_{h'}^t|x_{h'}^t)\widetilde{\ell}_{h'}^t(x_{h'}^t,a_{h'}^t) + \mathrm{KL}\big(\mu_{h'}\|\mu_{h'}^t\big)(x_{h'}^t)\Big)$$

$$= \sum_{h'=1}^{h} \mu_{1:h'-1}(x_{h'}^t)\Big(\eta\mu_{h'}(a_{h'}^t|x_{h'}^t)\widetilde{\ell}_{h'}^t(x_{h'}^t,a_{h'}^t) + \mathrm{KL}\big(\mu_{h'}\|\mu_{h'}^t\big)(x_{h'}^t)\Big)$$

$$\quad + \sum_{h'=h+1}^{H} \mu_{1:h'-1}(x_{h'}^t)\big(\mu_{h'}(a_{h'}^t|x_{h'}^t)\log Z_{h'+1}^t - \log Z_{h'}^t\big)$$

$$= \sum_{h'=1}^{h} \mu_{1:h'-1}(x_{h'}^t)\Big(\eta\mu_{h'}(a_{h'}^t|x_{h'}^t)\widetilde{\ell}_{h'}^t(x_{h'}^t,a_{h'}^t) + \mathrm{KL}\big(\mu_{h'}\|\mu_{h'}^t\big)(x_{h'}^t)\Big) - \mu_{1:h}(x_{h+1}^t)\log Z_{h+1}^t$$

$$= \sum_{h'=1}^{h-1} \mu_{1:h'-1}(x_{h'}^t)\Big(\eta\mu_{h'}(a_{h'}^t|x_{h'}^t)\widetilde{\ell}_{h'}^t(x_{h'}^t,a_{h'}^t) + \mathrm{KL}\big(\mu_{h'}\|\mu_{h'}^t\big)(x_{h'}^t)\Big)$$

$$\quad + \mu_{1:h-1}(x_h^t)\Big(\mu_h(a_h^t|x_h^t)\Big(\eta\widetilde{\ell}_h^t(x_h^t,a_h^t) - \log Z_{h+1}^t\Big) + \mathrm{KL}\big(\mu_h\|\mu_h^t\big)(x_h^t)\Big).$$

Therefore we deduce that

$$\mu_h^{t+1}(a_h|x_h^t) = \mu_h^t(a_h|x_h^t) \exp\Big(\mathbb{I}_{\{a_h=a_h^t\}}\Big(-\eta\widetilde{\ell}_h^t(x_h^t,a_h) + \log Z_{h+1}^t\Big) - \log Z_h^t\Big).$$

This concludes the proof.

## D  Efficient Computation of the Average Policy

In this appendix we explain how to efficiently compute the average policy in Theorem 1.

We define $\tau_h^t : \mathcal{X} \to \{0\} \cup \mathbb{N}$ by

$$\tau_h^t(x) := \max\big(\{0\} \cup \{1 \le k < t : x_h^k = x, \ k \in \mathbb{N}\}\big).$$

In other words, $\tau_h^t(x)$ is an index of an episode at which $x$ has been visited last time before $t$ (if it has been visited, otherwise returns 0). Further we define $\mathring{\mu}_{1:h}^t : \mathcal{X}_h \times \mathcal{A} \to [0, \infty)$ for each $h \in [H]$ by

$$\mathring{\mu}_{1:h}^t(x_h, a_h) := \sum_{u=1}^{t} \mu_{1:h}^u(x_h, a_h).$$

Using this function, we can compute the average policy since for any $t$

$$\frac{\sum_{u=1}^{t} \mu_{1:h}^u(x_h, a_h)}{\sum_{u=1}^{t} \mu_{1:h-1}^u(x_h)} = \frac{\mathring{\mu}_{1:h}^t(x_h, a_h)}{\sum_{a_h' \in \mathcal{A}} \mathring{\mu}_{1:h}^t(x_h, a_h')}.$$

Hence, we can compute the average policy after learning by using $\mathring{\mu}_{1:h}^T$.

Interestingly $\mathring{\mu}_{1:h}^t(x_h, a_h)$ can be computed while traversing a game tree by only using $\mu^t$ and a value available at the last time visitation to $x_h$. To see this, consider a fixed $(x_h, a_h) \in \mathcal{X}_h \times \mathcal{A}$ with $h > 1$ and let $\tau := \tau_h^t(x_h)$ for brevity. Since the policy does not change between $\tau + 1$ and $t$, we have that

$$\begin{aligned}
\mathring{\mu}_{1:h}^t(x_h, a_h) &= \sum_{u=1}^{t} \mu_{1:h}^u(x_h, a_h) \\
&= \sum_{u=1}^{\tau} \mu_{1:h}^u(x_h, a_h) + \sum_{u=\tau+1}^{t} \mu_{1:h-1}^u(x_h) \underbrace{\mu_h^u(a_h|x_h)}_{=\mu_h^t(a_h|x_h)} \\
&= \mathring{\mu}_{1:h}^\tau(x_h, a_h) + \left( \sum_{u=\tau+1}^{t} \mu_{1:h-1}^u(x_h) \right) \mu_h^t(a_h|x_h) \\
&= \mathring{\mu}_{1:h}^\tau(x_h, a_h) + \big( \mathring{\mu}_{1:h-1}^t(x_{h-1}, a_{h-1}) - \mathring{\mu}_{1:h-1}^\tau(x_{h-1}, a_{h-1}) \big) \mu_h^t(a_h|x_h),
\end{aligned}$$

where $(x_{h-1}, a_{h-1})$ is a unique predecessor of $x_h$. Therefore we can compute the average policy while traversing a game tree by using $\mu^t$ and $\mathring{\mu}_{1:h-1}^\tau(x_{h-1}, a_{h-1})$ stored at the last-visitation to $x_h$. For $h = 1$, a similar result holds by reading $\mu_{1:h-1}^u(x_h)$ as 1.

Therefore, once the learning ends, we can compute $\mathring{\mu}_{1:h}^T(x_h, a_h)$ for all visited information sets and actions, using stored transition data and $\mathring{\mu}_{1:h}^\tau$. (At non-visited information sets, the average policy chooses actions uniformly, and thus, no computation is required.) For a full pseudocode, see Algorithm 3.

# E  Additional proofs

We start with the proof of Lemma 2 stating that the losses are bounded.

*Proof of Lemma 2.* By definition, the reward function $r_h$ is a mapping from $\mathcal{S} \times \mathcal{A} \mapsto [0, 1]$. (To lift this assumption, one can simply scale all regret by the scalar.) Therefor it holds

$$0 \le \ell_h^t(x_h, a_h) \le \sum_{s_h \in x_h, b_h \in \mathcal{B}} p_{1:h}(s_h) \nu_{1:h}^t(s_h, b_h) \,.$$

From the tree structure and perfect recall, for any max-player policy $\mu$, $p_{1:h}(s_h)\nu_{1:h}^t(s_h, b_h)\mu_{1:h}(s_h, a_h)$ is a probability distribution over $\mathcal{S}_h \times \mathcal{A} \times \mathcal{B}$. Accordingly its partial sum is in $[0,1]$,

$$\sum_{s_h \in x_h, b_h \in \mathcal{B}} p_{1:h}(s_h)\nu_{1:h}^t(s_h, b_h)\mu_{1:h}(s_h, a_h) = \mu_{1:h}(x_h, a_h) \sum_{s_h \in x_h, b_h \in \mathcal{B}} p_{1:h}(s_h)\nu_{1:h}^t(s_h, b_h) \in [0, 1] \,.$$

Let us choose to be the policy that tries to reach $(x_h, a_h)$. (Recall that there is a unique action history up to $(x_h, a_h)$, so $\mu_{1:h}(x_h, a_h) = 1$.) It allows us to conclude

$$\mu_{1:h}(x_h, a_h) \sum_{s_h \in x_h, b_h \in \mathcal{B}} p_{1:h}(s_h)\nu_{1:h}^t(s_h, b_h) = \sum_{s_h \in x_h, b_h \in \mathcal{B}} p_{1:h}(s_h)\nu_{1:h}^t(s_h, b_h) \in [0, 1] \,.$$

$\square$

We are now ready to prove Lemma 4 equivalent in our setting to Lemma 1 by Neu (2015).

*Proof of Lemma 4.* Let $\widehat{\ell}_h^t(x_h, a_h)$ be the unbiased importance-sampling estimate of $\ell_h^t(x_h, a_h)$ defined in Equation 4. Then for any $t \in [T]$, $x_h \in \mathcal{X}_h$, and $a_h \in \mathcal{A}$,

$$
\begin{aligned}
\widetilde{\ell}_h^t(x_h, a_h) &= \frac{1 - r_h^t}{\mu_{1:h}^t(x_h, a_h) + \gamma} \mathbb{I}_{\{x_h = x_h^t, a_h = a_h^t\}} \\
&\leq \frac{1 - r_h^t}{\mu_{1:h}^t(x_h, a_h) + \gamma(1 - r_h^t)} \mathbb{I}_{\{x_h = x_h^t, a_h = a_h^t\}} \\
&= \frac{1}{\beta} \frac{\beta(1 - r_h^t) \mathbb{I}_{\{x_h = x_h^t, a_h = a_h^t\}} / \mu_{1:h}^t(x_h, a_h)}{1 + \gamma(1 - r_h^t) \mathbb{I}_{\{x_h = x_h^t, a_h = a_h^t\}} / \mu_{1:h}^t(x_h, a_h)} = \frac{1}{\beta} \frac{\beta \widehat{\ell}_h^t(x_h, a_h)}{1 + \beta \widehat{\ell}_h^t(x_h, a_h)/2} \\
&\leq \frac{1}{\beta} \log\left(1 + \beta \widehat{\ell}_h^t(x_h, a_h)\right),
\end{aligned}
$$

where $\beta := 2\gamma$, and the last inequality follows from $\dfrac{z}{1 + z/2} \leq \log(1 + z)$ for any $z \in [0, \infty)$.

Let $\widetilde{\lambda}^t := \sum_{x_h \in \mathcal{X}_h, a_h \in \mathcal{A}} \alpha^t(x_h, a_h) \widetilde{\ell}_h^t(x_h, a_h)$ and $\lambda^t := \sum_{x_h \in \mathcal{X}_h, a_h \in \mathcal{A}} \alpha^t(x_h, a_h) \ell_h^t(x_h, a_h)$. Note that we want to show $\sum_{t=1}^T (\widetilde{\lambda}^t - \lambda^t) \leq \log(1/\delta)$. Using the above inequality, we deduce that

$$
\begin{aligned}
\mathbb{E}^{t-1}\left[\exp\left(\widetilde{\lambda}^t\right)\right] &\leq \mathbb{E}^{t-1}\left[\exp\left(\sum_{x_h \in \mathcal{X}_h, a_h \in \mathcal{A}} \frac{\alpha^t(x_h, a_h)}{\beta} \log\left(1 + \beta \widehat{\ell}_h^t(x_h, a_h)\right)\right)\right] \\
&\leq \mathbb{E}^{t-1}\left[\prod_{x_h \in \mathcal{X}_h, a_h \in \mathcal{A}} \left(1 + \alpha^t(x_h, a_h) \widehat{\ell}_h^t(x_h, a_h)\right)\right] \\
&\leq \mathbb{E}^{t-1}\left[1 + \sum_{x_h \in \mathcal{X}_h, a_h \in \mathcal{A}} \alpha^t(x_h, a_h) \widehat{\ell}_h^t(x_h, a_h)\right] \\
&= 1 + \sum_{x_h \in \mathcal{X}_h, a_h \in \mathcal{A}} \alpha^t(x_h, a_h) \ell_h^t(x_h, a_h) \\
&\leq \exp\left(\sum_{x_h \in \mathcal{X}_h, a_h \in \mathcal{A}} \alpha^t(x_h, a_h) \ell_h^t(x_h, a_h)\right) = \exp\left(\lambda^t\right),
\end{aligned}
$$

where the second line follows from $z \log(1 + z') \leq \log(1 + z z')$ for any $z \in [0, 1]$ and $z' \in (-1, \infty)$, the third line follows from $\widehat{\ell}_h^t(x_h, a_h) \widehat{\ell}_h^t(x_h', a_h') = 0$ for any $(x_h, a_h) \neq (x_h', a_h')$, and the last line follows from $1 + z \leq \exp(z)$ for any $z \in \mathbb{R}$.

Define $Z_t := \exp(\widetilde{\lambda}^t - \lambda^t)$ and $M_t := \prod_{u=1}^t Z_u$. From the above inequality, we have that $\mathbb{E}[M_t] = \mathbb{E}\left[\mathbb{E}^{t-1}[M_t]\right] = \mathbb{E}\left[M_{t-1}\mathbb{E}^{t-1}[Z_t]\right] \leq \mathbb{E}[M_{t-1}] \leq \cdots \leq 1$. As a result, Markov's inequality implies

$$
\mathbf{Pr}\left(\sum_{t=1}^T (\widetilde{\lambda}^t - \lambda^t) \geq \log\frac{1}{\delta}\right) = \mathbf{Pr}\left(\log M_T \geq \log\frac{1}{\delta}\right) = \mathbf{Pr}(M_T \delta \geq 1) \leq \mathbb{E}[M_T]\delta \leq \delta.
$$

This concludes the proof. $\qquad\square$

We prove that the regularizer used by `IXOMD` is the Bregman divergence induced by the dilated entropy function, with uniform weights, introduced by Kroer et al. (2015). The dilated entropy function is defined by

$$
\Phi(\mu) = \sum_{h=1}^H \sum_{x_h \in \mathcal{X}_h, a_h \in \mathcal{A}_h} \mu_{1:h}(x_h, a_h) \log\left(\frac{\mu_{1:h}(x_h, a_h)}{\mu_{1:h}(x_h)}\right)
$$

where we denote $\mu_{1:h}(x_h) := \sum_{a \in \mathcal{A}_h} \mu_{1:h}(x_h, a)$.

**Lemma 9.** D *is the Bregman divergence associated to* $\Phi$.

*Proof.* First note that for any realization plan $\mu$ the gradient of $\Phi$ at $\mu$ is

$$\nabla_{h,x_h,a_h}\Phi(\mu) = \log\left(\frac{\mu_{1:h}(x_h, a_h)}{\mu_{1:h}(x_h)}\right) + 1 - \sum_{a\in\mathcal{A}_h}\frac{\mu_{1:h}(x_h, a)}{\mu_{1:h}(x_h)} = \log\left(\frac{\mu_{1:h}(x_h, a_h)}{\mu_{1:h}(x_h)}\right) .$$

It remains to conclude with

$$
\begin{aligned}
D_\Phi(\mu\|\mu') &= \Phi(\mu) - \Phi(\mu') - \langle\nabla\Phi(\mu'), \mu - \mu'\rangle \\
&= \sum_{h=1}^{H}\sum_{x_h\in\mathcal{X}_h, a_h\in\mathcal{A}_h}\mu_{1:h}(x_h, a_h)\log\left(\frac{\mu_{1:h}(x_h, a_h)}{\mu_{1:h}(x_h)}\right) \\
&\quad - \sum_{h=1}^{H}\sum_{x_h\in\mathcal{X}_h, a_h\in\mathcal{A}_h}\mu'_{1:h}(x_h, a_h)\log\left(\frac{\mu'_{1:h}(x_h, a_h)}{\mu'_{1:h}(x_h)}\right) \\
&\quad - \sum_{h=1}^{H}\sum_{x_h\in\mathcal{X}_h, a_h\in\mathcal{A}_h}\left(\mu_{1:h}(x_h, a_h) - \mu'_{1:h}(x_h, a_h)\right)\log\left(\frac{\mu'_{1:h}(x_h, a_h)}{\mu'_{1:h}(x_h)}\right) \\
&= \sum_{h=1}^{H}\sum_{x_h\in\mathcal{X}_h, a_h\in\mathcal{A}_h}\mu_{1:h}(x_h, a_h)\log\left(\frac{\mu_h(a_h|x_h)}{\mu'_h(a_h|x_h)}\right) \\
&= D(\mu\|\mu').
\end{aligned}
$$

$\square$

# F  Practical Implementation of `IXOMD`

In this appendix we provide a pseudocode for `PracticalIXOMD`, a practical version of `IXOMD`. Without loss of generality, we assume that $\mathcal{A} = \{1, \ldots, A\}$. We use Python-like list `List`, dictionary `Dict`, and Set `Set` objects (but we assume that the index of a list starts from 1). We also follow Python-like notations.

Algorithm 2 is a pseudocode for a memory-efficient implementation of the policy. It only stores action probabilities for observed information sets. We note that `MaxPlayerPolicy.batchUpdate`, which is called once per episode, has $\mathcal{O}(HA)$ time-complexity.

Algorithm 3 is a pseudocode for `PracticalIXOMD`. Line 6 to 28 correspond to the learning of the policy. As noted in the last paragraph, `MaxPlayerPolicy.batchUpdate` is called once per episode, and thus, the total time-complexity for the learning of the policy is $\mathcal{O}(THA)$. While traversing a game tree, we also perform the update of $\mathring{\mu}^t$, which is used to compute the average policy as described in Appendix D. For one traversal, this update requires $\mathcal{O}(THA)$ time-complexity in total. Line 29 to the end of the code correspond to the computation of $\overline{\mu}$ defined in Theorem 1. This part has $\mathcal{O}(\min(TH, X)A)$ time-complexity. As for the space-complexity, `muDot` requires the largest memory space, which is $\mathcal{O}(\min(TH, X)A)$.

---

**Algorithm 2:** `MaxPlayerPolicy`

---

1 **function** `__init__`():
2     `knownObs` = `Set`().
3     `actionProbas` = `Dict`().

4 ————————————————————————

5 **function** `getActionProba`($x, a$):
6     $p$ = `actionProbas`$[(x, a)]$ if $x$ in `knownObs` else $1/A$.
7     **return** $p$.

8 ————————————————————————

9 **function** `getActionProbas`($x$):
10     `probas` = `List`().
11     **for** $a = 1, \ldots, A$ **do**
12        `probas.append(getActionProba`($x, a$)).
13     **end**
14     **return** `probas`

15 ————————————————————————

16 **function** `update`($x, a, p$):
17     `actionProbas`$[(x, a)] = p$.
18     `knownObs.add`($x$).

19 ————————————————————————

20 **function** `batchUpdate(traj)`:
21     $\mu_{1:0} = 1$.
22     **for** $h = 1, \ldots, H$ **do**
23        $x_h, a_h, r_h$ = `traj`$[h]$.
24        $\mu_h$ = `actionProbas`$[(x_h, a_h)]$.
25        $\mu_{1:h} = \mu_{1:h-1}\mu_h$.
26     **end**
27     $Z_{H+1} = 1$.
28     **for** $h = H, \ldots, 1$ **do**
29        $x_h, a_h, r_h$ = `traj`$[h]$.
30        $\widetilde{\ell}_h = (1 - r_h)/(\mu_{1:h} + \gamma)$.
31        $Z_h = 1 - \mu_h + \mu_h \exp(-\eta\widetilde{\ell}_h + \log Z_{h+1})$.
32        `probas` = `getActionProbas`($x_h$).
33        **for** $a = 1, \ldots, A$ **do**
34           `update`($x_h, a$, `probas`$[a] \exp(\mathbb{I}_{a=a_h}(-\eta\widetilde{\ell}_h + \log Z_{h+1}) - \log Z_h))$.
35        **end**
36     **end**

---

**Algorithm 3:** `PracticalIXOMD` for the Max-player

**Input:** IX hyper-parameter $\gamma \in (0, \infty)$ and `OMD`'s learning rate $\eta \in (0, \infty)$.
**Output:** A near-NE policy for the max-player.

```
 1  pred = List(), muDot = List(), lastMuDotX = List().
 2  for h = 1, ..., H do
       // This initialization can be done later while playing the game.
 3     pred.append(Dict()), muDot.append(Dict()), lastMuDotX.append(Dict()).
 4  end
 5  policy = MaxPlayerPolicy(), lastIdx = Dict(), knownObs = Set().
    // Learn policies playing the game.
 6  for t = 1, ..., T - 1 do
```
7      $\texttt{traj} = \texttt{List}()$, $x_0^t = \varnothing$, $a_0^t = \varnothing$.

8      **for** $h = 1, \ldots, H$ **do**

9        Observe $x_h^t$ and compute $\texttt{probas} = \texttt{policy.getActionProbas}(x_h^t)$.

10        Execute $a_h^t$ sampled from $\texttt{probas}$, receive $r_h^t$, and $\texttt{traj.append}((x_h^t, a_h^t, r_h^t))$.

11        **if** $x_h^t \notin \texttt{knownObs}$ **then**

12          **for** $a = 1, \ldots, A$ **do**

13            $\texttt{muDot}[h][(x_h^t, a)] = 0$.

14          **end**

15          $\texttt{lastMuDotX}[h][x_h^t] = 0$, $\texttt{pred}[h][x_h^t] = (x_{h-1}^t, a_{h-1}^t)$, $\texttt{knownObs.add}(x_h^t)$.

16        **end**

17        **if** $h = 1$ **then**

18          $\texttt{diff} = t - \texttt{lastMuDotX}[h][x_h^t]$, $\texttt{lastMuDotX}[h][x_h^t] = t$.

19        **else**

20          $\texttt{diff} = \texttt{muDot}[h-1][(x_{h-1}^t, a_{h-1}^t)] - \texttt{lastMuDotX}[h][x_h^t]$.

21          $\texttt{lastMuDotX}[h][x_h^t] = \texttt{muDot}[h-1][(x_{h-1}^t, a_{h-1}^t)]$.

22        **end**

23        **for** $a = 1, \ldots, A$ **do**

24          $\texttt{muDot}[h][(x_h^t, a)] \mathrel{+}= \texttt{diff} \times \texttt{probas}[a]$.

25        **end**

26      **end**

27      $\texttt{policy.batchUpdate}(\texttt{traj})$.

28 **end**

// Compute the average policy.

29 $\texttt{averagePolicy} = \texttt{MaxPlayerPolicy}()$.

30 **for** $h = 1, \ldots, H$ **do**

     // Size of $\texttt{pred}[h].\texttt{keys}()$ is $\min(T, |\mathcal{X}_h|)$.

31      **for** $x_h \in \texttt{pred}[h].\texttt{keys}()$ **do**

32        $x_{h-1}, a_{h-1} = \texttt{pred}[h][x_h]$.

33        **if** $h = 1$ **then**

34          $\texttt{diff} = T - \texttt{lastMuDotX}[h][x_h]$.

35        **else**

36          $\texttt{diff} = \texttt{muDot}[h-1][(x_{h-1}, a_{h-1})] - \texttt{lastMuDotX}[h][x_h]$.

37        **end**

38        **for** $a = 1, \ldots, A$ **do**

39          $\texttt{muDot}[h][(x_h, a)] \mathrel{+}= \texttt{diff} \times \texttt{probas}[a]$.

40        **end**

41        $\texttt{sum} = \sum_{a' \in \mathcal{A}} \texttt{muDot}[h][(x_h, a')]$.

42        **for** $a = 1, \ldots, A$ **do**

43          $p = \texttt{muDot}[h][(x_h, a)]/\texttt{sum}$.

44          $\texttt{averagePolicy.update}(x_h, a, p)$.

45        **end**

46      **end**

47 **end**

48 **return** *Policy* `averagePolicy` *the average* $\overline{\mu}$ *of* $\mu^1, \ldots, \mu^T$ *defined in Theorem 1.*