# OpenReview forum: "Learning in two-player zero-sum partially observable Markov games with perfect recall"
_NeurIPS.cc/2021/Conference — NeurIPS 2021 Poster_

### Official Review · Reviewer_5YpW · 2021-06-29

**Rating:** 6
**Confidence:** 3

**Summary:**

This paper studies self-play algorithms in extensive form games with imperfect information. The proposed algorithm, IXOMD, can achieve the best-known sample complexity and achieve several desirable properties.

**Limitations And Societal Impact:**

This part is good.

**Main Review:**

I have a few concerns about the results:

1. How is the POMG setting in this paper related to EGII? Are they just one thing? Because the name of POMG always reminds me of general MG, where you don't have a tree structure.

2. The idea of reduce the problem into online linear bandit optimization and use a different divergence are brilliant. My understanding is that they are borrowed from existing literature and this paper improves them to high probability regret bound, right? Then I think maybe the authors want to make this explicit and add some explanation. For example, I don't understand why we need dilated entropy instead of KL as in adversarial MDP?

3. On time complexity. The authors claim the proposed algorithm requires only O(HA) time per episode. However, there are many existing works on computational barriers in POMDP, which is a special case of POMG. For example, see the discussion in 1.1 in [2]. Why this is not a contradiction?

4. On related work. The authors have made a long remark on related work on adversarial MDP. Actually, I think a more related setting is self-play algorithms in MG (with perfect information). Comparing with the sample complexity of algorithms for MG can also highlight the unique challenge in POMG. For example. see [3][4][5]

5. On the setting. In the related work section, the authors make the difference between full information and bandit information. So in the bandit feedback setting, can we see the state (instead of just the observation)? I know the algorithm proposed in this paper doesn't need this additional information, just a question concerning the setup. Furthermore, a natural question will be, it seems knowing the state when rolling out a trajectory doesn't help to achieve a better rate (by comparing the rate with existing works). Am I understanding correctly?

[1]  Regret Minimization in Games with Incomplete Information

[2] Sample-Efficient Reinforcement Learning of Undercomplete POMDPs

[3] Learning Zero-Sum Simultaneous-Move Markov Games Using Function Approximation and Correlated Equilibrium

[4] Provable Self-Play Algorithms for Competitive Reinforcement Learning

[5] Near-Optimal Reinforcement Learning with Self-Play

**Time Spent Reviewing:**

1

---

> ### Author Response · Authors · 2021-08-10
> **Author Response**
>
> Thank you very much for the useful comments and suggestions. We reply to specific points in the reviews below.
>
> > How is the POMG setting in this paper related to EGII? Are they just one thing? Because the name of POMG always reminds me of general MG, where you don't have a tree structure.
>
> Answer 11: In general, the POMG and EGII are not the same, because the POMG does not assume a tree structure while the EGII does. However, we assumed the POMG with a tree structure and the observation-generating mechanism explained in the paper. Then, it is possible to show that its special case is the EGII with the perfect-recall assumption. The main reason for using the equivalent POMG is to use notations more familiar to RL audiences.
>
> To see this, recall that both players simultaneously take actions in the POMG, whereas only one player does in the EGII. However, by picking up one player at each node and constraining its actions to have no effect, we effectively allow only one player to take an action at each node. (Note that the chance player can be assimilated to the state transition dynamics.) When all actions of a player have no effect at a particular node, any policy can do nothing there, and the player’s regret at the node is 0. As a result, all of our theoretical results hold without any modification.
>
> > The idea of reduce the problem into online linear bandit optimization and use a different divergence are brilliant. My understanding is that they are borrowed from existing literature and this paper improves them to high probability regret bound, right? Then I think maybe the authors want to make this explicit and add some explanation. For example, I don't understand why we need dilated entropy instead of KL as in adversarial MDP?
>
> Answer 12: Yes, you are right. The conversion of an extensive-form game to a linear program through the sequence-form representation is a well-known technique (cf. Full feedback paragraph of Introduction). It was first proposed in Romanovsky (1962) and rediscovered by von Stengel (1996), see also more recently Farina et al. (2020). We will be more explicit about this point.
>
> As for the choice of the dilated entropy distance-generating function, it allows an efficient dynamic-programming-like update as in Algorithm 1. Please refer to Line 93 to 96 and references therein for more details. Note that the dilated entropy can also be seen as a Kullback-Leibler divergence between the distribution of “sampled” realization plans.
>
> > On time complexity. The authors claim the proposed algorithm requires only O(HA) time per episode. However, there are many existing works on computational barriers in POMDP, which is a special case of POMG. For example, see the discussion in 1.1 in [2]. Why this is not a contradiction?
>
> Answer 13: We assume a tree structure and perfect recall, which are standard in game theory. As shown in [2], under an appropriate assumption, the POMDP is solvable. Therefore, it is not a contradiction even if we can efficiently compute a (near-) Nash equilibrium under our assumption.
>
> > On related work. The authors have made a long remark on related work on adversarial MDP. Actually, I think a more related setting is self-play algorithms in MG (with perfect information). Comparing with the sample complexity of algorithms for MG can also highlight the unique challenge in POMG. For example, see [3][4][5]
>
> Answer 14: In Introduction, we mainly discussed related works on solving the EGII and adversarial EGII, not adversarial MDP, because the focus of the paper is how to solve the EGII. We are sorry for the confusion.  Note that our setting and the one of [3][4][5] are not comparable thus the sample complexity neither (see Answer 4 for Reviewer BNiZ).
>
> > On the setting. In the related work section, the authors make the difference between full information and bandit information. So in the bandit feedback setting, can we see the state (instead of just the observation)? I know the algorithm proposed in this paper doesn't need this additional information, just a question concerning the setup.
>
> Answer 15: No, players have no access to the state. In the bandit feedback setting, players may observe only realizations of the game, i.e., observations, actions, and rewards encountered when playing games.
>
> > Furthermore, a natural question will be, it seems knowing the state when rolling out a trajectory doesn't help to achieve a better rate (by comparing the rate with existing works). Am I understanding correctly?
>
> Answer 16: It is a very interesting question. It might be possible to achieve a tighter regret bound when the state is observable. However, note that the learning of a quantity involving states (for example, the state-transition dynamics) typically has the sample complexity proportional to at least the square root of $S$. Because the number of states in an EGII is usually much larger than the number of observations, the learning of such a quantity must lead to a much worse bound than the one in the paper.

---

> > ### Comment · Reviewer_5YpW · 2021-08-18
> > **One more issue**
> >
> > Thanks for your detailed explanation. I have only one more technical question: it has been used several times in the paper that the loss function defined on page 6 is uniformly bounded by 1. Why this is the case? I check the appendix but didn't find any proof on this.

---

> > > ### Author Response · Authors · 2021-08-19
> > > **Re: One more issue**
> > >
> > > Thank you for the question. It holds as proven below. We will add this proof in the appendix.
> > >
> > > By definition, the reward function $r_h$ is a mapping from $\mathcal{S}_h \times \mathcal{A} \times \mathcal{B}$ to $[0, 1]$. (To lift this assumption, one can simply scale all regret by the scalar.) Therefore,
> > >
> > > $$
> > > 0 \leq \ell_h^t (x_h, a_h) \leq \sum_{s_h \in x_h, b_h \in \mathcal{B}} p_{1:h} (s_h) \nu_{1:h}^t (s_h, b_h).
> > > $$
> > >
> > > From the tree structure and perfect recall, $p_{1:h} (s_h) \nu_{1:h}^t (s_h, b_h) \mu_{1:h}^t (s_h, a_h)$ is a probability distribution over $\mathcal{S}_h \times \mathcal{A} \times \mathcal{B}$. Accordingly, its partial sum
> > >
> > > $$
> > > \sum_{s_h \in x_h, b_h \in \mathcal{B}} p_{1:h} (s_h) \nu_{1:h}^t (s_h, b_h) \mu_{1:h}^t (s_h, a_h)
> > > = \mu_{1:h}^t (x_h, a_h) \sum_{s_h \in x_h, b_h \in \mathcal{B}} p_{1:h} (s_h) \nu_{1:h}^t (s_h, b_h)
> > > $$
> > >
> > > is in $[0, 1]$. This holds for any max-player's policy $\mu$.
> > >
> > > Let us choose $\mu$ to be the policy that tries to reach $(x_h, a_h)$. (Recall that there is a unique action history up to $(x_h, a_h)$, so $\mu_{1:h}^t (x_h, a_h)=1$.) Then, it can be seen that
> > >
> > > $$
> > > \mu_{1:h}^t (x_h, a_h) \sum_{s_h \in x_h, b_h \in \mathcal{B}} p_{1:h} (s_h) \nu_{1:h}^t (s_h, b_h)
> > > = \sum_{s_h \in x_h, b_h \in \mathcal{B}} p_{1:h} (s_h) \nu_{1:h}^t (s_h, b_h) \in [0, 1].
> > > $$
> > >
> > > This proves that the loss is in $[0, 1]$.

---

> > > > ### Comment · Reviewer_5YpW · 2021-08-19
> > > > **comment on further response**
> > > >
> > > > Thanks for your explaination. I have now updated my score accordingly.

---

### Official Review · Reviewer_o4zK · 2021-07-15

**Rating:** 7
**Confidence:** 4

**Summary:**

This paper proposes Implicit Exploration Online Mirror Descent (IXOMD) algorithm that achieves sublinear regret in unknown extensive-form two-player zero-sum games with high probability. The setup is that player learn rewards only when actions are taken at some step, and do not learn about counterfactual reward they could have earned, have they made a different choice in the iteration. IXOMD is an adaptation of OMD, it uses importance sampling estimator of the losses with implicit exploration (to appropriately weigh the reward of the action taken and reduce its possibly high variance) and dilated entropy distance-generating function as a regularizer for the computation of next-step strategies.
Authors prove that IXOMD achieves sublinear regret with appropriate setting of the algorithm hyperparameters - the learning rate and the exploration penalty. Thus the algorithm can be used to approximate Nash equilibria (if both players use the algorithm) with high probability.


**Limitations And Societal Impact:**

The time horizon should be discussed, see the main review.

**Main Review:**

The paper is quite dense in notation, but clear and easy to follow. The proposed OMD adaptation achieves a high-probability regret bound, compared to other OMD adaptations that achieve the bound only in expectation. As authors note, there exist several other algorithms for their proposed online setting, and while they claim to achieve a better convergence rate in theory, it is a bit shame that they did not compare them in experiments as well. Especially, when other works [1] found that in practice CFR-based methods are still faster than OMD, despite worse convergence rates of CFR. Adding the experiments would improve my rating.

I do not understand the classification of "adversarial game" in Table 1. Some algorithms were classified as incapable of playing in the setting where the adversary also picks the game to play. This is a trivial game transformation, that can be formally achieved as the opponent has an additional decision node in which it picks the game to play, and thus this is a well specified large game (that can be decomposed into the multiple sub-games). Perhaps authors meant the adversary can pick any game whatsoever, from an infinite number of games? But wouldn't such an unlimited choice make the convergence rates vacuous, as they all depend on the number of infosets (that would be infinite as well?)

Authors write that "IXOMD requires almost no prior knowledge of the game. [..] We only require an oracle providing the possible actions at encountered information sets and a bound on A, B, and H to tune optimally the learning rate".
However, in Remark 3, to set the learning rate and exploration, also the time horizon T is used. Without the knowledge of the time horizon the regret bound can still contain the bias caused by the exploration. This should be mentioned as another (theoretical) limitation when this algorithm can be used. In constrast, for example [2,3] have worse convergence rates, but their algorithm is independent of the time horizon, because the hyperparameters do not depend on time, thus may be suitable in a repeated play where we do not know the time horizon.

Other comments:
- L37-38 while CFR+ achieves 1/sqrt(T) in theory, in practice it is often 1/T and it does not need a choice of hyperparameters.


[1]  Optimistic Regret Minimization for Extensive-Form Games via Dilated Distance-Generating Functions

[2]  Model-Free Online Learning in Unknown Sequential Decision Making Problems and Games.

[3]   Stochastic Regret Minimization in Extensive-Form Games.

**Time Spent Reviewing:**

6

---

> ### Author Response · Authors · 2021-08-10
> **Author Response**
>
> Thank you very much for the detailed and critical comments. We reply to specific points in the reviews below.
>
> > ...it is a bit shame that they did not compare them in experiments as well. Especially, when other works [1] found that in practice CFR-based methods are still faster than OMD, despite worse convergence rates of CFR.
>
> Answer 8: We agree that it is an important future work to compare IXOMD to CFR-based methods. That being said, we believe the paper provides a solid and sufficient theoretical contribution that there is a statistically and computationally efficient model-free algorithm having $\tilde{\mathcal{O}}(1/\sqrt{T})$ convergence rate with neither balanced strategy nor prior on the game.
>
> > I do not understand the classification of "adversarial game" in Table 1. Some algorithms were classified as incapable of playing in the setting where the adversary also picks the game to play.
>
> Answer 9: We are sorry for the confusing term. By “adversarial game”, we meant a game where the opponent may choose the reward function and state-transition dynamics, in addition to its strategy. Therefore, the information sets are unchanged. We will be more specific.
>
> > Authors write that "IXOMD requires almost no prior knowledge of the game. [..] We only require an oracle providing the possible actions at encountered information sets and a bound on A, B, and H to tune optimally the learning rate". However, in Remark 3, to set the learning rate and exploration, also the time horizon T is used.
>
> Answer 10: It seems there is a misunderstanding. $T$ is not the time horizon but the number of game episodes. We would like to recall that our focus is how to learn a (near-) Nash equilibrium though self-play, not competing against the opponent. Therefore, a typical situation would be the one where it is allowed to run an algorithm for a finite budget of computational time. In this case, one would perform a preliminary experiment to see computational time per episode, and then carry out as many episodes as possible. In such a case, $T$ is known before the full experiment.
>
> Besides, one may use the doubling trick (Cesa-Bianch and Lugosi, 2006) and can transform IXOMD to an anytime algorithm. The doubling trick results in a regret bound larger than only by a constant (less than 4) factor, so the bound of anytime IXOMD is still better than those in [2, 3]. Another possibility is to use a $t$-dependent learning rate.
>
> We will add discussion paragraphs about these points.
>
> Reference:
>
> Nicolo Cesa-Bianchi and Gabor Lugosi. 2006. Prediction, Learning, and Games. Cambridge University Press, USA.

---

> > ### Comment · Reviewer_o4zK · 2021-08-31
> > **Response to authors.**
> >
> > Thank you for your explanations. I have no other concerns.

---

### Official Review · Reviewer_mGny · 2021-07-18

**Rating:** 7
**Confidence:** 4

**Summary:**

This paper studies the problem of learning Nash equilibrium policies in two-player zero-sum extensive games, under the assumption of bandit feedback. The authors propose an algorithm that is based on Implicit Exploration and OMD, and they provide an high-probability regret bound that scales sub-linearly in the number of iterations and polynomially in the size of the game. Interestingly, the regret bound holds even in an online setting (not in self-play) in which both the policies of the opponent and the game structure are chosen adversarially.

**Limitations And Societal Impact:**

Adequate.

**Main Review:**

Originality: The problem studied in the paper is not novel, but it recently received a lot of attention in the literature. The ideas used in the paper are novel; in particular that of using Implicit Exploration.

Quality: As far as I am concerned, the proofs are correct.

Clarity: Overall, the paper is clear, but there are some minor typos and I have some suggestions:
- Line 9: "Moreover..." => "Moreover,...".
- Line 19: "max..." => "max-...".
- Line 58: "consider" => "considers".
- Line 66: It is not clear to which algorithm the sentence "This algorithm..." refers to.
- Line 69: "such that" => "such as".
- Line 96: Is it "O(H)" or "O(H A)"?
- Line 120: "p_0" is missing from the definition.
- Line 128: "define" => "defined"; I think it's better to define "X_h" here.
- Section 2: The notation is cumbersome and hard to follow; I would suggest the authors trying to avoid the use of multiple subscripts and apices and that of many indexes, such as "h", "h'", and "k". (Don't know actually if notation can be improved easily).
- Line 184: "gives" => "give".
- Line 9 in Algorithm 1: "with..." is redundant.
Also, it seems to me that the algorithm is working over the sequence-form strategy space. Am I correct? If yes, why didn't you frame the whole technical contribution in the sequence-form setting? I think that would have helped clarifying notation and presentation.

Significance: The paper provides a good contribution to its field, and I think it fits well within the scope of the NeurIPS conference.

**Time Spent Reviewing:**

4

---

> ### Author Response · Authors · 2021-08-10
> **Author Response**
>
> Thank you very much for your kind comments and for pointing out typos. We promise to fix them. We reply to specific points in the reviews below.
>
> > The ideas used in the paper are novel; in particular that of using Implicit Exploration.
>
> Answer 5: Indeed, implicit exploration has been applied to various problem settings, but it has never been applied to the EGII setting. We believe that it is a novel finding and idea that the combination of implicit exploration and the perfect-recall assumption results in a simple, fast, and computationally efficient algorithm.
>
> > Line 96: Is it "O(H)" or "O(H A)"?
>
> Answer 6: It is $\mathcal{O}(HA)$ since we need to update probabilities of all actions at each time step in an episode. If we wrote that it is $\mathcal{O}(H)$ somewhere else in the paper, it is a typo, so we would be very happy to fix it.
>
> > Also, it seems to me that the algorithm is working over the sequence-form strategy space. Am I correct? If yes, why didn't you frame the whole technical contribution in the sequence-form setting? I think that would have helped clarifying notation and presentation.
>
> Answer 7: Yes, it is indeed working over the sequence-form strategy space. However, as we presented in Algorithm 1, efficient dynamic-programming-like implementation is possible thanks to the dilated entropy distance-generating function. If we were to use the sequence-form setting throughout the paper, the presentation of Algorithm 1 would need the introduction of non-sequence-form notations. Accordingly, there seems to be a trade-off between the clarity of presenting the algorithm and theoretical results. We will think of a better presentation.

---

### Official Review · Reviewer_BNiZ · 2021-07-26

**Rating:** 7
**Confidence:** 3

**Summary:**

This paper studies partially observable extensive games with perfect recall. They propose a computationally efficient model-free algorithm with $\sqrt{T}$-regret. The algorithm works by reducing the tree-structured game to a linear bandits problem and then applying some variant of EXP-IX (similar to Jin et al., 2020) to solve it.


**Main Review:**

This paper is well-written and easy to follow. The new algorithm looks computationally efficient and has regret guarantee matching the best known result. Moreover, it can also play favourably against adversarial components.

Is the conversion to online linear bandits first proposed in this work? If not, maybe it would be better to point this out explicitly in section 3.1. Besides, the algorithm designed in this paper bears much similarity to Jin et al., 2020. I feel it’s worthwhile to point it out in the paper.

Could you provide a direct comparison between this paper and Lanctot et al. (2009); Farina et al. (2020) in terms of computational complexity? I believe this comparison is necessary especially when these works have the same regret.

How do you compare this work to [1]? I think it would be interesting to compare the two settings studied in this paper and [1]. Which setting is more general or practical?

Overall this is an interesting paper making reasonable contributions.

[1] A Sharp Analysis of Model-based Reinforcement Learning with Self-Play.
Qinghua Liu, Tiancheng Yu, Yu Bai, Chi Jin. ICML 2021



**Time Spent Reviewing:**

3

---

> ### Author Response · Authors · 2021-08-10
> **Author Response**
>
> Thank you very much for the valuable comments and suggestions. We reply to specific points in the reviews below.
>
> > Is the conversion to online linear bandits first proposed in this work? If not, maybe it would be better to point this out explicitly in section 3.1.
>
> Answer 1: No, it is not our contribution (cf. Full feedback paragraph of Introduction). The conversion of an extensive-form game to a linear program through the sequence-form representation is a well-known technique. It was first proposed in Romanovsky (1962) and rediscovered by von Stengel (1996). See also the more recent work by Farina et al. (2020) We will be more explicit about this point in Section 3.1.
>
> > Besides, the algorithm designed in this paper bears much similarity to Jin et al., 2020. I feel it’s worthwhile to point it out in the paper.
>
> Answer 2: On a high level, our algorithm may resemble that of Jin et al. (2020). However, their algorithm cannot be applied to our setting, to the best of our knowledge. Particularly, because their algorithm is model-based, states must be known to the players. As a result, it is unclear how to handle the imperfect information setting. Furthermore, it is unclear how to handle the non-stationary state-transition dynamics due to updates of the opponent's strategy. Another critical difference is the choice of the regularizer: dilated entropy vs unnormalized KL, the former of which leads to simple and computationally friendly updates in our case.
>
> > Could you provide a direct comparison between this paper and Lanctot et al. (2009); Farina et al. (2020) in terms of computational complexity? I believe this comparison is necessary especially when these works have the same regret.
>
> Answer 3: The complexity of the outcome-sampling algorithm using the CFR algorithm as regret minimizer is the same as our algorithm IXOMD as explained in Section 4.2 by Farina et al. (2020). Note that, to our knowledge, they did not explain how to compute the average policy, but the method we provided in appendix C can be combined with the outcome sampling CFR algorithm. However, this is not a full comparison, because it is unclear how to know and compute the exploration profile (cf. Bandit feedback,model-free).
>
> > How do you compare this work to [1]? I think it would be interesting to compare the two settings studied in this paper and [1]. Which setting is more general or practical?
>
> Answer 4: The setting considered in [1] is the perfect information setting, whereas our setting is the imperfect information one. By setting each information set of both players to be a singleton of the state, the perfect information setting is recovered. However, we assume perfect recall and the tree structure of the game. Although those assumptions are standard in game theory, they make the direct comparison of our setting and theirs impossible.

---

> > ### Comment · Reviewer_BNiZ · 2021-08-23
> > **re: author response**
> >
> > Thanks for the response. The authors have addressed all my questions. This is a nice paper and I vote for acceptance.

---

### Decision · Program_Chairs · 2021-09-27

**Decision:**

Accept (Poster)

**Comment:**

This paper studies partially observable extensive games with perfect recall. They propose a computationally efficient model-free algorithm with sqrt(T) regret. All reviewers believe the results are solid and strong, and the techniques used in this paper are interesting. Therefore, we recommend acceptance.